# ON THE INHERENT VULNERABILITY OF RANDOMIZED MODELS TO NAGGING ATTACK

## ABSTRACT

There has been a long debate on whether inference-time randomness enhances or reduces the safety of deep learning models. In this paper, we formalize a realistic threat model, and provide both theoretical and empirical evidence demonstrating that randomness in inference undermines model safety. The threat we consider is *nagging attack*, where an adversary repeatedly queries the model with the same adversarial example, exploiting the model's inherent randomness until a failure occurs. Theoretically, we show that nagging attack introduces a fundamental vulnerability for randomized models, which cannot be eliminated by refining their design. To address this challenge, we propose *seed-rotation*, a deployment strategy that de-randomizes the model within fixed intervals while retaining stochasticity across intervals. We prove that seed-rotation makes the randomized model safer. Empirically, we compared seed-rotation to the standard randomized deployment across multiple randomized defense methods. The results consistently showed that seed-rotation achieved higher adversarial robustness than the randomized deployment.

## 1 INTRODUCTION

Recently, there is a growing trend towards the incorporation of randomness in deep learning models, particularly during inference. Examples include Large Language Models (LLMs) that leverage sampling-based approaches (*e.g.*, temperature sampling) (Brown et al., 2020; Holtzman et al., 2020; Wiher et al., 2022), diffusion models (Ho et al., 2020; Song et al., 2021a; Rombach et al., 2022; Song et al., 2021b), and auto-regressive image generation models (Li et al., 2024a; Yu et al., 2024; Ren et al., 2024; 2025). These models utilize inherent randomness to introduce diversity in their outputs or to enhance generative capabilities.

From a security perspective, there has been a long debate on whether randomness increases the safety of models (Gnecco Heredia et al., 2023; Pinot et al., 2019; 2020; 2022; Meunier et al., 2021) or decreases that (Lucas et al., 2023; Gao et al., 2022; Dbouk & Shanbhag, 2022; Lee & Kim, 2023; Wang et al., 2024). On one hand, randomness can provide a certificate on the robust radius around a data point (Cohen et al., 2019; Salman et al., 2020; Gnecco Heredia et al., 2023). On the other hand, it might expose new vulnerabilities that adversaries could exploit (Lucas et al., 2023; Gao et al., 2022; Dbouk & Shanbhag, 2022). This debate raises an important question: *Does the incorporation of randomness ultimately enhance model security, or does it increase susceptibility to attacks?*

To answer this question, we first need to establish a realistic threat model. Consider a scenario where a malicious user manipulates a model to generate harmful outputs, such as instructions for illegal activities (Zou et al., 2023; Wei et al., 2023; Li et al., 2024b; Carlini et al., 2023a; Qi et al., 2024). For randomized models, a simple yet particularly effective strategy is the nagging attack (Wang et al., 2024; Lucas et al., 2023), in which the adversary repeatedly queries the model with the same input, exploiting its inherent randomness until a failure occurs. For example, consider a commercial LLM deployed with stochastic decoding and accessible through a cloud-based API. Although the system is designed to reject malicious prompts, an attacker could craft a weak jailbreak prompt that succeeds with probability $0.1$. By simply resubmitting this prompt, the attacker would, on average, obtain a successful jailbreak within ten queries, potentially eliciting harmful outputs such as detailed bomb-making instructions (Zou et al., 2023).

While this vulnerability of randomness has been observed, prior work only examined specific empirical defenses (Wang et al., 2024; Lucas et al., 2023), and lacked a general understanding or a practical security strategy to address that beyond advocating for deterministic defenses (Lucas et al., 2023). To formally investigate this vulnerability of randomized models, we adopt a well-established threat model for adversarial attacks on image classification models. We begin by defining and analyzing the adversarial nagging risk (ANR) that randomized models face. We then show that *whenever the adversary is allowed more than one query, all nontrivial randomized models incur an ANR strictly higher than the previously considered adversarial risk.*

Instead of advocating for the elimination of randomness in model design for safety reasons, we explore the potential of a different deployment strategy for randomized models to enhance safety. Drawing inspiration from cryptography, we propose *seed-rotation*, where the random seed used at inference is fixed within a given time interval (*e.g.*, one day) but periodically refreshed. Within each interval, the model behaves deterministically, while across intervals it retains the appearance of a randomized model. Unlike a permanently deterministic model, which can eventually be compromised and treated as white-box, seed-rotation ensures that the adversary does not know the active seed. From the adversary's perspective, the model outputs remain indistinguishable from samples of a randomized defense. Importantly, seed-rotation is a deployment strategy and does not require modifications to the underlying randomized model or defense. We prove that under seed-rotation, the worst-case risk does not exceed the adversarial risk of the randomized deployment. Furthermore, we conducted numerical experiments to assess the effectiveness of seed-rotation on various randomized defense methods. The empirical results showed that all randomized defense methods exhibited higher accuracy against adversarial attacks with seed-rotation than with randomized deployment.

Our main contributions are as follows:

- We formalize the risk of nagging attack, denoted as ANR, and show that whenever the adversary is allowed more than one query, all nontrivial randomized models incur an ANR strictly higher than the previously considered adversarial risk.
- We propose the seed-rotation deployment strategy for randomized models, and show that this approach provides enhanced security compared to standard randomized deployment.
- We conducted numerical experiments to evaluate various randomized defense methods and verified that they all had higher accuracy with seed-rotation than with randomized deployment when defending against adversarial attacks.

## 2 RELATED WORK

**General theory on randomized defense.** There has been an ongoing debate on whether randomness increases the safety of models (Gnecco Heredia et al., 2023; Pinot et al., 2019; 2020; 2022; Meunier et al., 2021) or decreases that (Lucas et al., 2023; Gao et al., 2022; Dbouk & Shanbhag, 2022; Lee & Kim, 2023; Wang et al., 2024).

Proponents of randomized defenses argue that carefully designed randomized models outperform deterministic models. Gnecco Heredia et al. (2023) identified the condition for a probabilistic classifier to outperform deterministic ones. Pinot et al. (2019) showed that noise drawn from the Exponential family guaranteed a good accuracy against adversarial attack. Pinot et al. (2020) demonstrated that any deterministic classifier could be outperformed by a randomized one from a game-theoretic perspective. Meunier et al. (2021) designed an algorithm to learn a mixture of a finite number of classifiers to approximate a Nash equilibrium. Pinot et al. (2022) introduced an efficient noise injection method to build robust randomized classifiers.

In contrast, critics argue that most existing randomized defenses fail under stronger or adaptive attacks. Gao et al. (2022) found that many randomized defense designs lacked enough randomization to even defend against standard PGD (Madry et al., 2018) attack. Dbouk & Shanbhag (2022) showed that current randomized defense designs were broken by a stronger attack adaptively designed for randomized defenses. Lucas et al. (2023) argued that deterministic defenses were similarly robust to randomized defenses. Recent evaluations (Lee & Kim, 2023; Wang et al., 2024; Li et al., 2025) showed that the robustness of DiffPure (Nie et al., 2022), a popular randomized defense, had been over-estimated. Specifically, empirical results (Wang et al., 2024) showed that the robustness of DiffPure decreased a lot against nagging attack.

**Empirical and certified randomized defense methods.** A variety of methods incorporate randomness to defend against adversarial attacks (Cohen et al., 2019; Salman et al., 2019; 2020; Carlini et al., 2023b; Nie et al., 2022; Chen et al., 2024; Li et al., 2025). One major line of work (Cohen et al., 2019; Salman et al., 2019; 2020; Carlini et al., 2023b) was based on Randomized Smoothing (RS) (Lecuyer et al., 2019; Cohen et al., 2019), which provided certified robustness guarantees. RS constructs a smoothed classifier by adding Gaussian noise to inputs, evaluates multiple noise-corrupted inputs, and uses majority voting to predict the label. If the vote margin is large enough, it can certify robustness within a certain $l_2$-bounded region. Follow-up work (Salman et al., 2019; 2020; Carlini et al., 2023b) improved the underlying base classifiers for better noise robustness without altering the RS framework. One work was Denoised Smoothing (DS) (Salman et al., 2020), which trained a denoiser to denoise the Gaussian noise added by RS framework, and thereby reduced the cost of training each classifier to classify Gaussian noise-corrupted inputs. Another line of research (Nie et al., 2022; Chen et al., 2024; Li et al., 2025) focused on empirical robustness using diffusion models. Although these models did not explicitly rely on theoretical guarantees with randomness, they required stochastic sampling processes – typically involving adding Gaussian noise and iterative denoising – which inherently introduced randomness into the defense pipeline. The standard method among them was DiffPure (Nie et al., 2022), which added Gaussian noise to the input, and used the denoising capability of diffusion models to purify the input from a mixture of Gaussian and adversarial noise.

## 3 PRELIMINARY

**Threat model.** We assume the adversary has full access to the architecture, parameters, and defense mechanism of the target model. However, for the randomized model, the attacker has no access to the random seed or the internal random state of the model. The ultimate goal of the adversary is to break a remote deployment of the model. To break the remote deployment, the adversary is allowed only black-box access: one may submit inputs and observe final outputs, without access to intermediate variables, gradients, or internal states. The adversary is also allowed to construct and query a local substitute model. The adversary is allowed to query it with unlimited query budget. When querying the local substitute, the adversary has full gradient access for white-box optimization. In this paper, we make a strong-adversary assumption that the local substitute shares the same parameters and architecture as the remote model. To summarize, we assume a gray-box threat model, where the adversary has white-box access to a local substitute, but has to compromise the black-box remote deployment.

**Adversarial risk of a classifier.** We begin by formalizing the notion of adversarial risk. Our notation follows a previous work (Gnecco Heredia et al., 2023), restated here for clarity and completeness.

Let $\mathcal{X}$ denote the input space and $\mathcal{Y}$ the label space, with $|\mathcal{Y}| < \infty$, where $|\cdot|$ denotes the number of elements in a set. We assume a data distribution $\rho \in \mathcal{P}(\mathcal{X} \times \mathcal{Y})$ over input-label pairs $(x, y) \in \mathcal{X} \times \mathcal{Y}$.

A *deterministic classifier* is a mapping $h : \mathcal{X} \to \mathcal{Y}$, which assigns a single predicted label to each input. The standard 0-1 loss, or accuracy, is defined as:

$$l^{0\text{-}1}((x, y), h) = \mathbb{1}\{h(x) \neq y\}, \tag{1}$$

where $(x, y) \in \mathcal{X} \times \mathcal{Y}$ is a data point and $\mathbb{1}\{\cdot\}$ is the indicator function.

A *probabilistic classifier* is a mapping $\mathbf{h} : \mathcal{X} \to \mathcal{P}(\mathcal{Y})$, which outputs a distribution over labels for each input. In the case of $K$-class classification ($|\mathcal{Y}| = K$), $\mathbf{h}(x)$ can be viewed as a $K$-dimensional probability vector, with the $i$-th element in the vector being the probability of being predicted as the $i$-th class. On each query, a predicted label $\hat{y} \sim \mathbf{h}(x)$ is sampled from the output distribution. The 0-1 loss of a probabilistic classifier is:

$$l^{0\text{-}1}((x, y), \mathbf{h}) = \mathbb{E}_{\hat{y} \sim \mathbf{h}(x)}[\mathbb{1}\{\hat{y} \neq y\}] = 1 - \mathbf{h}(x)_y, \tag{2}$$

where $\mathbf{h}(x)_y$ denotes the probability assigned to the true label $y$.

Let $B(x) \subseteq \mathcal{X}$ denote the allowed perturbation set around an input $x \in \mathcal{X}$. Formally, $B : \mathcal{X} \to 2^{\mathcal{X}}$ maps each input to a subset of the input space, specifying valid perturbations. An adversarial attack seeks an input $x' \in B(x)$ that maximizes the chance of misclassification. That is, the attacker attempts to find a point within the perturbation set that causes the classifier to produce an incorrect

output. In practice, many prior works (Goodfellow et al., 2015; Madry et al., 2018; Carlini & Wagner, 2017; Gnecco Heredia et al., 2023) instantiate the perturbation set using norm balls around the input:

$$B_\epsilon(x) := \{x' \in \mathcal{X} : d(x, x') \le \epsilon\}, \tag{3}$$

where $d$ is a distance metric on $\mathcal{X}$, and $\epsilon \ge 0$ controls the perturbation budget.

We define the *adversarial risk* of a classifier as the expected worst-case loss under allowable perturbations. For a probabilistic classifier $\mathbf{h}$ and a deterministic one $h$, the adversarial risks are:

$$\mathcal{R}_\epsilon(\mathbf{h}) := \mathbb{E}_{(x,y)\sim\rho}\left[\sup_{x' \in B_\epsilon(x)} l^{0\text{-}1}((x', y), \mathbf{h})\right], \quad \mathcal{R}_\epsilon(h) := \mathbb{E}_{(x,y)\sim\rho}\left[\sup_{x' \in B_\epsilon(x)} l^{0\text{-}1}((x', y), h)\right]. \tag{4}$$

**Probabilistic classifier from a base hypothesis set.** In practice, a probabilistic classifier $\mathbf{h}$ outputs a predicted label $\hat{y} \in \mathcal{Y}$ for each query using random numbers during inference. This behavior can be interpreted as selecting a deterministic classifier $h \in \mathcal{H}_b$, drawn from a distribution over a set of hypotheses. We refer to this hypothesis set as the *base hypothesis set* (BHS), denoted as $\mathcal{H}_b$. Formally, let $\mu \in \mathcal{P}(\mathcal{H}_b)$ be a probability measure over the BHS $\mathcal{H}_b$. A probabilistic classifier $\mathbf{h}_\mu$ is then defined as:

$$\mathbf{h}_\mu(x)_y := \mathbb{P}_{h\sim\mu}(h(x) = y), \ \forall x \in \mathcal{X}, y \in \mathcal{Y}. \tag{5}$$

If the BHS of a probabilistic classifier $h_\mu$ is finite ($|\mathcal{H}_b| < \infty$), then the classifier is denoted as a *randomized ensemble classifier* (REC) (Gnecco Heredia et al., 2023).

## 4 PROBABILISTIC CLASSIFIERS INEVITABLY SUFFER FROM NAGGING RISK

We formalize the ANR, which quantifies the vulnerability of probabilistic classifiers under nagging attacks (Section 4.1). We then prove that whenever the adversary is allowed more than one query, all nontrivial randomized models inevitably suffer from an ANR higher than the standard adversarial risk (Section 4.2). Then we provide some simple yet useful properties of ANR (Section 4.3). Moreover, we provide an estimate of the number of queries needed for nagging attack (Section 4.4). Proofs to all theorems are provided in Appendix D.

### 4.1 DEFINITION OF NAGGING RISK

Nagging attack means that the attacker is able to query the model for multiple times to get one successful attack. We start from defining the risk against nagging attack on probabilistic classifiers.

**Definition 4.1** (Adversarial nagging risk of probabilistic classifier). The adversarial risk of a probabilistic classifier $\mathbf{h}$ against nagging attack with query budget $n$ is

$$\mathcal{R}_\epsilon^{(n)}(\mathbf{h}) := \mathbb{E}_{(x,y)\sim\rho}\left(\sup_{x' \in B_\epsilon(x)}\left(1 - [\mathbf{h}(x')_y]\right)^n\right). \tag{6}$$

From the definition, the adversarial risk of probabilistic classifier is a special case of ANR with $n = 1$. That is, $\mathcal{R}_\epsilon(\mathbf{h}) = \mathcal{R}_\epsilon^{(1)}(\mathbf{h})$. Practically, the nagging attack is executed by finding the adversarial input $x' \in B_\epsilon(x)$ that minimizes the predicted probability $\mathbf{h}(x')_y$, and then the attacker repeatedly queries the probabilistic classifier with the same adversarial input $x'$. This allows the attacker to exploit the probabilistic nature of the model, as each query outputs a potentially different predicted label $\hat{y} \sim \mathbf{h}(x')$. Then we recall the definition of *maximal simultaneous vulnerability* $\mu^{\max}$.

**Definition 4.2** (Maximal simultaneous vulnerability (Gnecco Heredia et al., 2023)). Let $\mathcal{H}_{vb}(x, y) \subseteq \mathcal{H}_b$ denote the *vulnerable subset* and $\mu^{\max}(x, y)$ the *maximal simultaneous vulnerability*, which indicates the maximal proportion of models that a single adversarial example breaks, defined below.

$$\mathcal{H}_{vb}(x, y) = \{h \in \mathcal{H}_b : \exists x'_h \in B_\epsilon(x) \text{ such that } h(x'_h) \ne y\}, \tag{7}$$

$$\mathfrak{H}_{svb}(x, y) = \{\mathcal{H}' \subseteq \mathcal{H}_b : \exists x' \in B_\epsilon(x) \text{ such that } \forall h \in \mathcal{H}', h(x') \ne y\}, \tag{8}$$

$$\mu^{\max}(x, y) = \sup_{\mathcal{H}' \in \mathfrak{H}_{svb}(x,y)} \mu(\mathcal{H}'). \tag{9}$$

We further introduce non-degenerating probabilistic classifiers, whose set of adversarial examples has non-zero measure.

**Definition 4.3** (Non-degeneracy). Let $\mathbf{h}_\mu$ be a probabilistic classifier. It is called a *non-degenerating probabilistic classifier* on data distribution $\rho \in \mathcal{P}(\mathcal{X} \times \mathcal{Y})$ if it satisfies:

$$\mu[\mathcal{H}_{vb}(x, y)] > 0 \Rightarrow \mu^{\max}(x, y) > 0 \text{ almost surely.} \tag{10}$$

This condition is purely technical and does not exclude widely used models. For example, an REC is always a non-degenerating probabilistic classifier (see Appendix A for details).

### 4.2 Probabilistic Classifiers Inevitably Suffer from Nagging Risk

A probabilistic classifier is said to be *nontrivial* if it outperforms all deterministic classifiers in its BHS $\mathcal{H}_b$. This condition is formalized as $\mathcal{R}_\epsilon(\mathbf{h}_\mu) < \inf_{h \in \mathcal{H}_b} \mathcal{R}_\epsilon(h)$ (Gnecco Heredia et al., 2023). We show that a nontrivial probabilistic classifier inevitably incurs an ANR that exceeds its original adversarial risk.

**Theorem 4.1.** *Let $\mathbf{h}_\mu$ be a probabilistic classifier over BHS $\mathcal{H}_b$. If $\mathcal{R}_\epsilon(\mathbf{h}_\mu) < \inf_{h \in \mathcal{H}_b} \mathcal{R}_\epsilon(h)$, then for all $n \in \mathbb{N}^*$ with $n > 1$,*

$$\mathcal{R}_\epsilon^{(n)}(\mathbf{h}_\mu) > \mathcal{R}_\epsilon(\mathbf{h}_\mu). \tag{11}$$

The theorem implies that once the adversary is permitted more than a single query, the additional adversarial risk is non-negligible. The nagging attack is straightforward to execute: after crafting a strong adversarial example against the randomized model, the adversary can simply resubmit the same input repeatedly, relying on the model's inherent randomness to induce the risk.

### 4.3 Properties of Nagging Risk

We hereby give some simple properties of nagging risk. We begin with the upper and lower bounds, then analyze the limit of nagging risk when the query budget is infinite.

**Theorem 4.2.** *Let $\mathbf{h}_\mu$ be a probabilistic classifier. Then for all $n \in \mathbb{N}^*$,*

$$\mathcal{R}_\epsilon(\mathbf{h}_\mu) \leq \mathcal{R}_\epsilon^{(n)}(\mathbf{h}_\mu) \leq 1 - (1 - \mathcal{R}_\epsilon(\mathbf{h}_\mu))^n. \tag{12}$$

By further analyzing Definition 4.1, we observe the shortcomings of probabilistic classifiers with respect to ANR. As the number of queries $n \to +\infty$, we find that the adversarial input $x'$ succeeds in breaking the classifier if and only if $\mathbf{h}(x')_y < 1$. This insight is formalized in Appendix D (Lemma D.5). Furthermore, with non-degeneracy, we convert the theorem into an existence problem, which makes the calculation feasible.

**Theorem 4.3.** *Let $\mathbf{h}_\mu$ be a non-degenerating probabilistic classifier. Then,*

$$\lim_{n \to +\infty} \mathcal{R}_\epsilon^{(n)}(\mathbf{h}_\mu) = \mathbb{P}_{(x,y) \sim \rho}(\exists x' \in B_\epsilon(x), h \in \mathcal{H}_b \text{ such that } h(x') \neq y). \tag{13}$$

Hence, we can equivalently write the limit of adversarial nagging risk as the probability of the existence of a single adversarial input–classifier pair $(x', h)$ that causes misclassification. This reformulation offers a more feasible route to empirical estimation: one can optimize over both the input and the random number to identify such a non-vanishing failure case, making the limiting risk computable in practice.

### 4.4 Estimating Queries Needed for Nagging Attack

In this section, we provide an estimation of the number of queries needed for nagging attack to guarantee an ANR $\mathcal{R}_\epsilon^{(n)}(\mathbf{h}_\mu) \geq 1 - \delta$, where $\delta$ is a pre-determined error rate.

**Theorem 4.4.** *Let $\mathbf{h}_\mu$ be an REC on $M$ base hypotheses $h_1, h_2, \ldots, h_M$ with uniform prior $\mu(h_i) = \frac{1}{M}$, $i = 1, 2, \ldots, M$. Let $(x, y)$ be the data point that we focus on. Assume for each base hypothesis $h_i$, there exists a disjoint-vulnerable adversarial example $x'_i$, s.t., $\forall j \neq i$, $h_i(x'_i) \neq y$, $h_j(x'_i) = y$,*

*i.e., the adversarial example is specific to that hypothesis. Assume the pairwise Kullback–Leibler divergence is uniformly bounded:*

$$D_{KL}(P_{i,x'} \| P_{j,x'}) \leq d_{\max} < \infty, \; \forall i \neq j, x' \in B_{\epsilon}(x), \tag{14}$$

*where $P_{i,x'}$ represents the output probabilities – which can be known from logits – of model $h_i$ with regard to input $x'$. Then, for the specific data point $(x, y)$,*

$$\mathcal{R}_{\epsilon}^{(n)}(\mathbf{h}_{\mu}) \leq \frac{nd_{\max} + \log 2}{\log M}. \tag{15}$$

*Equivalently, to guarantee $\mathcal{R}_{\epsilon}^{(n)}(\mathbf{h}_{\mu}) \geq 1 - \delta$ one must have*

$$n \geq \frac{(1 - \delta) \log M - \log 2}{d_{\max}}. \tag{16}$$

The above theorem provides a practical estimator for the queries needed for nagging attack, which is an indicator of the safety level. Practically, the values of $d_{\max}$ and $M$ can be estimated through experiments. $P_{i,x'}, P_{j,x'}$ are taken from white-box access to model logits, and $d_{\max}$ is estimated by Monte-Carlo sampling of $x'$. $M$ represents the randomness and $\log M$ is proportional to the number of random bits used. Since all randomized models originate from the uniform random bits from the generator, this is a direct construction from practical models.

## 5 SEED-ROTATION DEPLOYMENT OF RANDOMIZED MODEL

The previous section showed that a randomized model inevitably incurs an adversarial nagging risk that exceeds its original adversarial risk due to its inherent randomness. Specifically, this randomness allows attackers to gradually exploit the model through nagging attack. To mitigate this vulnerability, we propose a *seed-rotation* deployment strategy that preserves the confusion effect of randomness for the attacker but makes the model behaviour deterministic for a certain interval.

### 5.1 METHOD

Seed-rotation is done by fixing the random seed for inference within a given time interval (*e.g.* one day) and periodically refreshing it. Within each interval, the model behaves deterministically, while across intervals it retains the appearance of a randomized model. Seed-rotation ensures that the adversary does not know the active seed. From the adversary's perspective, the model outputs remain indistinguishable from samples of a randomized defense. Seed-rotation prevents attackers from exploiting the model's stochasticity through nagging attack by de-randomizing the model during each interval. However, there is an additional risk introduced by the attacker's capability to exploit the now-deterministic behavior of the model, same as query-based black-box attack against deterministic model. Combining the mitigation of nagging risk and the introduction of risk against query attack, we analyze the overall risk of seed-rotation below.

### 5.2 THEORETICAL PERSPECTIVE

A seed-rotation classifier is an REC consisting of $M$ base hypotheses with the BHS $\mathcal{H}_b = \{h_1, h_2, \ldots, h_M\}$. During a given interval (*e.g.*, 1 day), the system deterministically uses a single model $h_i \in \mathcal{H}_b$, selected according to a probability distribution $\mathcal{P}_M(\mathcal{H}_b)$. Each such interval is referred to as a *seed rotation cycle*.

This deployment strategy thwarts nagging attacks. Since multiple queries within a seed rotation cycle are effective as only one query to the seed-frozen model, $n$ queries of nagging attack now span $n$ seed rotation cycles to achieve the same cumulative effect as before. The trade-off, however, is that each cycle exposes a deterministic model, making the system potentially vulnerable to query-based black-box attacks. We therefore analyze the robustness of seed-rotation under such attacks. Proofs to all theorems in this section are provided in Appendix D.

**Definition 5.1** (Query attack)**.** A black-box hard-label query-based attack (or *query attack*) $Q(\epsilon)$, parameterized by the perturbation radius $\epsilon$, is defined as a mapping $Q_{\epsilon} : \mathcal{X} \times \mathcal{Y} \times \mathcal{H} \times \mathbb{N}^* \to \mathcal{X}$, which generates an adversarial example for a given input, label, classifier, and query count.

The risk under query attack with query count $n$ is defined as:

$$\mathcal{R}_{Q(\epsilon)}^{(n)} := \mathbb{P}_{(x,y)\sim\rho}(h(Q_\epsilon(x, y, h, n)) \neq y), \tag{17}$$

which characterizes the adversarial vulnerability of a deterministic classifier to query-based attacks with at most $n$ queries to the model.

Comparing randomized deployment to seed-rotation, an asymptotic can be achieved when attackers have an unbounded query budget.

**Theorem 5.1.** *Let $\mathbf{h}_\mu$ be a non-degenerating probabilistic classifier, and let $Q$ be any query attack. Then,*

$$\lim_{n\to+\infty} \mathbb{E}_{h\sim\mu}\left[\mathcal{R}_{Q(\epsilon)}^{(n)}(h)\right] \leq \mathbb{E}_{h\sim\mu}[\mathcal{R}_\epsilon(h)] \leq \lim_{n\to+\infty} \mathcal{R}_\epsilon^{(n)}(\mathbf{h}_\mu). \tag{18}$$

Intuitively, this is because nagging to the limit exploit all non-zero probabilities to deterministically break the model in certain trial, whose risk is more than the average white-box risk of all base hypotheses, and is larger than the average black-box risk.

For evaluation, we ensure that the query attack is not too weak by a common technique of selecting the best adversarial example among queries. That ensures the *monotonic* property of attack. That is, for all $x \in \mathcal{X}, y \in \mathcal{Y}, h \in \mathcal{H}, m, n \in \mathbb{N}^*$ with $m < n$, the following holds:

$$h(Q_\epsilon(x, y, h, m)) \neq y \Rightarrow h(Q_\epsilon(x, y, h, n)) \neq y. \tag{19}$$

Moreover, we initialize the attack with the standard Expectation-over-Transformation (EoT) technique over a distribution of classifiers. EoT selects the best adversarial example on average. Then we use the query attack to refine the adversarial examples that fail on the currently seeded model. Initialization with EoT ensures a strong lower bound of adversarial risk, avoiding under-estimation of the adversarial risk of seed-rotation models (formalized in Theorem D.5 of Appendix D).

### 5.3 EVALUATION SETUP

In practical scenarios, the adversary's query budget is finite. In this finite regime, there is no theoretical guarantee favoring one deployment over the other. In particular, if the individual deterministic classifiers $h \in \mathcal{H}_b$ are not intrinsically robust in the white-box setting, the security of seed-rotation relies on the practical difficulty of query-based black-box attacks – lacking access to gradients and intermediate variables – to efficiently uncover adversarial examples under a limited query budget. This was empirically confirmed before, especially in the hard-label case (Chen & Gu, 2020). Hence, we conducted numerical experiments to empirically compare the two deployment strategies. Specifically, for a fixed query budget $n$, we evaluate the average risk of the seed-rotation deployment, given by $\mathbb{E}_{h\sim\mu}\left[\mathcal{R}_{Q(\epsilon)}^{(n)}(h)\right]$, and the risk of the randomized deployment, given by $\mathcal{R}_\epsilon^{(n)}(\mathbf{h}_\mu)$. This setup allows us to observe how the two methods perform under identical computational constraints.

**Evaluating randomized deployment.** To estimate the adversarial risk $\mathcal{R}_\epsilon^{(n)}(\mathbf{h}_\mu)$ of a randomized defense, we adopt the standard EoT technique (Athalye et al., 2018; Tramer et al., 2020; Croce & Hein, 2020) to craft adversarial examples. For each data point $(x, y) \sim \rho$, we craft an adversarial example $x'$ by approximately solving $x' \approx \arg\max_{x_1 \in B_\epsilon(x)} (1 - \mathbf{h}(x_1)_y)$ using gradient-based adversarial attack with EoT. Once $x'$ is generated, we sample $n$ independent models $h_1, h_2, \ldots, h_n \sim \mu$, and query each model on $x'$ to obtain predictions $\hat{y}_i = h_i(x')$. The pointwise risk for $(x, y)$ is then estimated by the event that any prediction is incorrect in the $n$-nagging attack, which is $\max_{1 \leq i \leq n} \mathbb{1}\{\hat{y}_i \neq y\}$. Averaging this quantity over multiple test examples gives an empirical estimate of $\mathcal{R}_\epsilon^{(n)}(\mathbf{h}_\mu)$.

**Evaluating seed-rotation deployment.** To evaluate the average risk of the seed-rotation deployment, we first sample a random seed for the model, representing the now-deterministic model $h_0 \sim \mu$. For each data point $(x, y) \sim \rho$, we begin by generating a universal adversarial example $x'_0$ using the same EoT-based attack applied to $\mathbf{h}_\mu$. We then query the model $h_0$ with $x'_0$. If $h_0(x'_0) \neq y$, the risk is marked as 1 for that data point and model. Otherwise, we invoke a monotonic query attack algorithm $Q_\epsilon$ with a query budget $n$, targeting $h_0$, to obtain a refined adversarial input $x'_n$. Then the risk is flagged as $\mathbb{1}\{h_0(x'_n) \neq y\}$. To estimate the overall risk, we repeat this procedure across multiple

Table 1: Robust accuracy (%) of ResNet-50 on CIFAR-10 under $l_\infty$-bounded attacks ($\epsilon = \frac{8}{255}$) for various randomized defenses. "Clean" indicates accuracy on unperturbed inputs. "EoT" refers to standard EoT attack, which is the same for randomized deployment and seed-rotation. For each query budget $n$, we report results for two deployment strategies: "Rand" for randomized and "Rot" for seed-rotation. "Diff" indicates the improvement of Rot over Rand.

| Defense | Clean | EoT | $n = 10$ | | | $n = 100$ | | | $n = 1000$ | | |
|---|---|---|---|---|---|---|---|---|---|---|---|
| | | | Rand | Rot | Diff | Rand | Rot | Diff | Rand | Rot | Diff |
| RN@0.25 | 82.5 | 32.4 | 14.0 | 32.1 | **+18.1** | 8.3 | 31.6 | **+23.3** | 6.4 | 28.5 | **+22.1** |
| RN@0.5 | 69.6 | 36.4 | 12.1 | 34.6 | **+22.5** | 4.5 | 31.9 | **+27.4** | 2.1 | 26.5 | **+24.4** |
| RN@1.0 | 51.4 | 32.6 | 6.4 | 26.3 | **+19.9** | 1.1 | 21.5 | **+20.4** | 0.3 | 17.3 | **+17.0** |
| RS$_{100}$@0.25 | 87.0 | 28.5 | 25.8 | 28.4 | **+2.6** | 24.3 | 28.4 | **+4.1** | 22.8 | 28.4 | **+5.6** |
| RS$_{100}$@0.5 | 76.6 | 35.9 | 31.7 | 35.9 | **+4.2** | 29.2 | 35.9 | **+6.7** | 27.3 | 35.9 | **+8.6** |
| RS$_{100}$@1.0 | 61.1 | 35.0 | 29.5 | 34.6 | **+5.1** | 26.1 | 34.6 | **+8.5** | 23.8 | 34.6 | **+10.8** |
| DS$_{100}$@0.25 | 83.3 | 17.8 | 15.3 | 17.9 | **+2.6** | 13.8 | 17.9 | **+4.1** | 12.5 | 17.9 | **+5.4** |
| DS$_{100}$@0.5 | 51.7 | 21.1 | 17.9 | 21.2 | **+3.3** | 15.9 | 21.2 | **+5.3** | 14.0 | 21.2 | **+7.2** |
| DS$_{100}$@1.0 | 12.4 | 10.3 | 9.9 | 10.3 | **+0.4** | 9.6 | 10.3 | **+0.7** | 9.4 | 10.3 | **+0.9** |
| DiffPure | 91.5 | 47.6 | 35.7 | 47.6 | **+11.9** | 31.9 | 47.4 | **+15.5** | 29.8 | 45.1 | **+15.3** |

independent samples $(x, y) \sim \rho$ and seeds $h_0 \sim \mu$, and take the average risk to empirically estimate $\mathbb{E}_{h \sim \mu}\left[\mathcal{R}_{Q(\epsilon)}^{(n)}(h)\right]$ for the seed-rotation deployment. Code-level implementation details on how to control the randomness in inference are provided in Appendix B.

# 6 EXPERIMENTAL RESULTS

In this section, we demonstrate the practical benefit of seed-rotation when compared to randomized deployment under the same query budget $n$.

**Experimental setup.** We evaluated randomized models on CIFAR-10 dataset (Krizhevsky et al., 2009) and ImageNet-1k dataset (Russakovsky et al., 2015). Due to the high query cost (up to 1000 queries per sample), we randomly selected a subset of 2000 examples from ImageNet-1k for evaluation. The CIFAR-10 models used were ResNet-50 and ResNet-101 (He et al., 2016). For ImageNet-1k, we used ResNet-50 and Swin-L (Liu et al., 2021). We assessed three standard randomized defenses: Randomized Smoothing (RS) (Cohen et al., 2019), Denoised Smoothing (DS) (Salman et al., 2020), and DiffPure (Nie et al., 2022). RS and DS were tested at noise magnitudes $\sigma \in \{0.25, 0.5, 1.0\}$ using 100 noise samples, denoted RS$_{100}$@$\sigma$. For RS, we used classifiers pretrained on Gaussian noise with the corresponding magnitude (Cohen et al., 2019). DS employed pretrained denoisers to denoise Gaussian noise (Salman et al., 2020). To highlight contrast, we introduced Random Noise (RN), a naive defense where Gaussian noise $\hat{\epsilon} \sim N(0, \sigma^2\mathbf{I})$ is added directly to the input, followed by classification on $x + \hat{\epsilon}$. RN corresponds to RS with a single sample. RN also used classifiers pretrained on noise. The EoT attacks were performed with 100 EoT iterations for RS and DS, and 20 EoT iterations for DiffPure and RN. The base attack for EoT attack was a reduced version of AutoAttack (Croce & Hein, 2020). Specifically, we applied PGD with cross-entropy loss (PGD-ce) (Madry et al., 2018) and PGD with DLR loss (PGD-dlr) (Croce & Hein, 2020), each run for 20 steps. The adversarial example with the highest cross-entropy loss was used. These same adversarial examples were reused to evaluate the adversarial nagging risk, as discussed in Section 5.3. For the seed-rotation deployment, we used RayS (Chen & Gu, 2020) as the query attack algorithm. We initialized the attack with the same EoT attacks as the attack to evaluate randomized deployment. We evaluated the models under $l_\infty$ or $l_2$ perturbation bound.

The results for ResNet-50 on CIFAR-10 under the $l_\infty$ perturbation bound are shown in Table 1. Across all randomized defenses, we observed a consistent decline in accuracy as the number of attack attempts $n$ increased, confirming that these models are vulnerable to nagging attacks. The accuracy degradation followed an approximately exponential trend, which is consistent with the theoretical prediction in Theorem 4.2. Among the evaluated defenses, RS and DS exhibited slower degradation of accuracy, suggesting they have lower intrinsic randomness. In contrast, RN and DiffPure suffered more rapid drops in accuracy, indicating higher susceptibility to exploitation by nagging attack. When

Table 2: Robust accuracy (%) of ResNet-50 on ImageNet-1k under $l_\infty$-bounded attacks ($\epsilon = \frac{4}{255}$) for various randomized defenses. Notations are consistent with Table 1.

| Defense | Clean | EoT | $n = 10$ | | | $n = 100$ | | | $n = 1000$ | | |
|---|---|---|---|---|---|---|---|---|---|---|---|
| | | | Rand | Rot | Diff | Rand | Rot | Diff | Rand | Rot | Diff |
| RN@0.25 | 67.5 | 7.1 | 4.1 | 7.2 | **+3.1** | 3.2 | 7.2 | **+4.0** | 2.7 | 7.2 | **+4.5** |
| RN@0.5 | 60.0 | 18.6 | 11.2 | 18.9 | **+7.7** | 8.4 | 18.8 | **+10.4** | 7.1 | 18.4 | **+11.3** |
| RN@1.0 | 45.1 | 23.5 | 12.8 | 21.6 | **+8.8** | 8.8 | 21.2 | **+12.4** | 6.7 | 19.2 | **+12.5** |
| RS$_{100}$@0.25 | 69.8 | 4.9 | 4.5 | 4.7 | **+0.2** | 4.2 | 4.7 | **+0.5** | 4.0 | 4.7 | **+0.7** |
| RS$_{100}$@0.5 | 62.7 | 15.1 | 14.0 | 15.3 | **+1.3** | 13.2 | 15.3 | **+2.1** | 13.0 | 15.3 | **+2.3** |
| RS$_{100}$@1.0 | 51.2 | 21.3 | 19.5 | 21.4 | **+1.9** | 18.8 | 21.4 | **+2.6** | 17.9 | 21.4 | **+3.5** |
| DS$_{100}$@0.25 | 65.4 | 9.2 | 8.2 | 9.1 | **+0.9** | 8.0 | 9.1 | **+1.1** | 7.9 | 9.1 | **+1.2** |
| DS$_{100}$@0.5 | 46.2 | 16.7 | 16.0 | 17.1 | **+1.1** | 15.3 | 17.1 | **+1.8** | 15.0 | 17.1 | **+2.1** |
| DS$_{100}$@1.0 | 13.7 | 7.9 | 7.3 | 7.8 | **+0.5** | 7.0 | 7.8 | **+0.8** | 6.9 | 7.8 | **+0.9** |
| DiffPure | 77.5 | 8.0 | 4.7 | 7.9 | **+3.2** | 3.9 | 7.9 | **+4.0** | 3.2 | 7.9 | **+4.7** |

comparing randomized deployment to seed-rotation under the same query budget $n$, we consistently found that seed-rotation achieved higher robustness. This demonstrates that under practical query constraints, seed-rotation mitigates the effects of nagging attacks. Results on the ImageNet-1k dataset (Table 2) support the same conclusions. Additional experiments conducted under $l_2$ norm bounds and other architectures on both datasets (see Appendix B) also revealed consistent patterns, further validating the generality of our findings across perturbation types, datasets, and models.

## 7 CONCLUSION AND DISCUSSION

In this paper, we demonstrate that nontrivial randomized models are inherently vulnerable to nagging attacks – a form of repeated-query adversarial attack that exploits randomness. To address this vulnerability, we propose seed-rotation for any randomized model, which freezes the random seed and periodically refreshes it. We show both theoretically and through empirical evaluations that seed-rotation consistently offers stronger robustness than the standard randomized deployment. Our findings provide a new pathway for safely deploying randomized models, without discouraging research into stochastic defenses.

We conclude by discussing key implications and extensions of this work.

**Implications of the inherent vulnerability of randomized models to nagging attacks.** We demonstrate that probabilistic classifier inevitably incurs an ANR that exceeds its original adversarial risk (Theorem 4.1). Notably, we compare two distinct risks of the same randomized models, while prior work compared randomized models to deterministic alternatives, and only analyzed the standard adversarial risk (Gnecco Heredia et al., 2023). Moreover, we assume an attacker with unlimited power, characterized by the definition of risks (Eqs. (4) and (6)), in contrast to an attacker with limited strength (Pinot et al., 2020).

**Extension beyond classifiers.** Our theoretical framework generalizes beyond classification tasks. By generalizing the prediction and evaluation mechanisms, the framework naturally applies to any model equipped with an evaluator function and a corresponding risk metric. Recall the jailbreak setting in introduction (Section 1). As long as we can craft a discriminator to decide whether the output is harmful or not, our framework extends naturally to that. This generalization is formalized in Appendix C.1.

**Extension to natural risk.** Our theoretical results also extend beyond adversarial robustness. Specifically, by setting the perturbation set $B(x) = \{x\}$ for all $x \in \mathcal{X}$, we reduce the problem to *natural risk*, *i.e.*, standard model evaluation. In the natural risk setting, since the only valid adversarial example is the original example $x$ itself, query attacks have no chance of increasing risk. In this setting, we can refer to Theorem 4.1 and get a stronger result: $\mathcal{R}^{(n)}(\mathbf{h}_\mu) > \mathbb{E}_{h \sim \mu}\left[\mathcal{R}_Q^{(n)}(h)\right], \ \forall n \in \mathbb{N}^*$, which shows that randomized models incur additional risk from nagging attack but seed-rotation models remain unaffected. Further details are provided in Appendix C.2.

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

# APPENDIX

## A  INSTANTIATIONS OF PROBABILISTIC CLASSIFIERS

There are three common types of probabilistic classifiers (Gnecco Heredia et al., 2023):

- *Randomized ensemble classifier (REC):* A finite mixture of $M$ classifiers $\{h_1, \ldots, h_M\}$, where $\mu$ is a probability distribution over these $M$ classifiers.

- *Weight-noise injection classifier (WNC):* Each hypothesis $h_w \in \mathcal{H}_b$ corresponds to a neural network parameterized by weights $w \in \mathcal{W} \subseteq \mathbb{R}^p$. Randomness enters via sampling $w \sim \mu$, where $\mu$ is a distribution over the weight space $\mathcal{W}$.

- *Input-noise injection classifier (INC):* Built from a fixed deterministic classifier $\tilde{h} : \mathcal{X} \to \mathcal{Y}$, with random noise injected into the input. Each hypothesis is defined as $h_\eta(x) = \tilde{h}(x + \eta)$, where the input perturbation $\eta \sim \mu$, and the BHS is $\mathcal{H}_b = \{h_\eta : \eta \in \mathcal{X}\}$.

We introduced the non-degenerating condition as defined in Definition 4.3. To demonstrate that this condition is purely technical and does not exclude widely used models, we show that standard classes of probabilistic classifiers satisfy it. Proofs to the following propositions are provided in Section D.

**Proposition A.1.** *An REC $\mathbf{h}_\mu$ is always a non-degenerating probabilistic classifier.*

*Remark* A.1. In practical settings, all probabilistic classifiers are implemented as RECs, since random number generators operate over finite state spaces, assigning non-zero probability to each instantiation.

**Proposition A.2.** *A WNC $\mathbf{h}_\mu$ with base mapping $h$ that is continuous on $\mathcal{W} \times \mathcal{X}$ is non-degenerating, provided that the adversarial input $x'$ misclassifies the data point $(x, y)$ in a non-vanishing way. That is, the total predicted probability of the incorrect labels remains strictly higher than the predicted probability of the true label almost surely.*

*Remark* A.2. An INC is a special case of a WNC. Given a deterministic base classifier $h_w$ and input noise $\eta \in \mathcal{X}$, an INC can be represented as a WNC with weights $(w, \eta) \in \{w\} \times \mathcal{X}$. The hypothesis is defined as $\tilde{h}_{(w,\eta)}(x) := h_w(x + \eta)$.

## B  ADDITIONAL NUMERICAL EXPERIMENTS

We provide additional numerical experiments on different datasets, model architectures and perturbation bounds, to verify the generality of our findings.

**Implementation details.**  We instantiated a dedicated random number generator (RNG) in PyTorch to produce all stochastic components during inference, such as Gaussian noise. For the randomized deployment, the RNG was seeded once at the beginning of the program to ensure reproducibility, with the seed drawn uniformly from the range $[0, 2^{32})$. In contrast, for seed-rotation, the RNG was re-seeded with the same value for every inference pass, ensuring deterministic behaviour per query. The seed was sampled uniformly from $[0, 2^{32})$ using a global RNG independent of the one used for inference-time randomness.

**Compute resources.**  All experiments were run on an NVIDIA RTX 3090 GPU. Evaluating RN with a ResNet-50 model on 2000 samples from the ImageNet-1k dataset required under 1 GPU hour. Evaluating $RS_{100}$ on the same dataset consumed 60 GPU hours, $DS_{100}$ required 100 GPU hours, and DiffPure required 168 GPU hours.

**Confidence intervals.**  We report binomial confidence intervals in Table S1, which are omitted from the main text due to table width limit. Other experimental results exhibit confidence intervals of similar magnitude.

**Additional results.**  We extended our evaluation to include the $l_2$ perturbation setting, under which both RS and DS provided certified robustness. Results for ResNet-50 on CIFAR-10 and ImageNet-1k under the $l_2$ threat model are presented in Tables S2 and S3, confirming conclusions consistent with

Table S1: Robust accuracy (%) with confidence intervals for results in Table 2. Binomial confidence intervals with 95 % confidence are marked at subscripts.

| Defense | Clean | EoT | $n = 10$ | | $n = 100$ | | $n = 1000$ | |
|---|---|---|---|---|---|---|---|---|
| | | | Rand | Rot | Rand | Rot | Rand | Rot |
| RN@0.25 | $67.5_{\pm 2.1}$ | $7.1_{\pm 1.1}$ | $4.1_{\pm 0.9}$ | $7.2_{\pm 1.1}$ | $3.2_{\pm 0.8}$ | $7.2_{\pm 1.1}$ | $2.7_{\pm 0.7}$ | $7.2_{\pm 1.1}$ |
| RN@0.5 | $60.0_{\pm 2.1}$ | $18.6_{\pm 1.7}$ | $11.2_{\pm 1.4}$ | $18.9_{\pm 1.7}$ | $8.4_{\pm 1.2}$ | $18.8_{\pm 1.7}$ | $7.1_{\pm 1.1}$ | $18.4_{\pm 1.7}$ |
| RN@1.0 | $45.1_{\pm 2.2}$ | $23.5_{\pm 1.9}$ | $12.8_{\pm 1.5}$ | $21.6_{\pm 1.8}$ | $8.8_{\pm 1.2}$ | $21.2_{\pm 1.8}$ | $6.7_{\pm 1.1}$ | $19.2_{\pm 1.7}$ |
| RS$_{100}$@0.25 | $69.8_{\pm 2.0}$ | $4.9_{\pm 0.9}$ | $4.5_{\pm 0.9}$ | $4.7_{\pm 0.9}$ | $4.2_{\pm 0.9}$ | $4.7_{\pm 0.9}$ | $4.0_{\pm 0.9}$ | $4.7_{\pm 0.9}$ |
| RS$_{100}$@0.5 | $62.7_{\pm 2.1}$ | $15.1_{\pm 1.6}$ | $14.0_{\pm 1.5}$ | $15.3_{\pm 1.6}$ | $13.2_{\pm 1.5}$ | $15.3_{\pm 1.6}$ | $13.0_{\pm 1.5}$ | $15.3_{\pm 1.6}$ |
| RS$_{100}$@1.0 | $51.2_{\pm 2.2}$ | $21.3_{\pm 1.8}$ | $19.5_{\pm 1.7}$ | $21.4_{\pm 1.8}$ | $18.8_{\pm 1.7}$ | $21.4_{\pm 1.8}$ | $17.9_{\pm 1.7}$ | $21.4_{\pm 1.8}$ |
| DS$_{100}$@0.25 | $65.4_{\pm 2.1}$ | $9.2_{\pm 1.3}$ | $8.2_{\pm 1.2}$ | $9.1_{\pm 1.3}$ | $8.0_{\pm 1.2}$ | $9.1_{\pm 1.3}$ | $7.9_{\pm 1.2}$ | $9.1_{\pm 1.3}$ |
| DS$_{100}$@0.5 | $46.2_{\pm 2.2}$ | $16.7_{\pm 1.6}$ | $16.0_{\pm 1.6}$ | $17.1_{\pm 1.7}$ | $15.3_{\pm 1.6}$ | $17.1_{\pm 1.7}$ | $15.0_{\pm 1.6}$ | $17.1_{\pm 1.7}$ |
| DS$_{100}$@1.0 | $13.7_{\pm 1.5}$ | $7.9_{\pm 1.2}$ | $7.3_{\pm 1.1}$ | $7.8_{\pm 1.2}$ | $7.0_{\pm 1.1}$ | $7.8_{\pm 1.2}$ | $6.9_{\pm 1.1}$ | $7.8_{\pm 1.2}$ |
| DiffPure | $77.5_{\pm 1.8}$ | $8.0_{\pm 1.2}$ | $4.7_{\pm 0.9}$ | $7.9_{\pm 1.2}$ | $3.9_{\pm 0.8}$ | $7.9_{\pm 1.2}$ | $3.2_{\pm 0.8}$ | $7.9_{\pm 1.2}$ |

Table S2: Robust accuracy (%) of ResNet-50 on ImageNet-1k under $l_2$-bounded attacks ($\epsilon = 2$) for various randomized defenses. Notations are consistent with Table 1.

| Defense | Clean | EoT | $n = 10$ | | | $n = 10^2$ | | | $n = 10^3$ | | |
|---|---|---|---|---|---|---|---|---|---|---|---|
| | | | Rand | Rot | Diff | Rand | Rot | Diff | Rand | Rot | Diff |
| RN@0.25 | 67.5 | 26.3 | 18.2 | 25.9 | **+7.7** | 15.5 | 25.9 | **+10.4** | 13.0 | 25.9 | **+12.9** |
| RN@0.5 | 60.0 | 33.9 | 23.0 | 33.8 | **+10.8** | 17.9 | 33.7 | **+15.8** | 15.1 | 33.3 | **+18.2** |
| RN@1.0 | 45.1 | 33.0 | 19.1 | 30.6 | **+11.5** | 13.3 | 30.2 | **+16.9** | 9.5 | 29.1 | **+19.6** |
| RS$_{100}$@0.25 | 69.8 | 23.2 | 22.4 | 23.4 | **+1.0** | 22.0 | 23.4 | **+1.4** | 21.6 | 23.4 | **+1.8** |
| RS$_{100}$@0.5 | 62.7 | 32.9 | 30.6 | 32.5 | **+1.9** | 29.5 | 32.5 | **+3.0** | 28.8 | 32.5 | **+3.7** |
| RS$_{100}$@1.0 | 51.2 | 31.3 | 29.1 | 30.9 | **+1.8** | 28.1 | 30.9 | **+2.8** | 27.4 | 30.9 | **+3.5** |
| DS$_{100}$@0.25 | 65.4 | 18.8 | 17.2 | 18.7 | **+1.5** | 16.8 | 18.7 | **+1.9** | 16.5 | 18.7 | **+2.2** |
| DS$_{100}$@0.5 | 46.2 | 10.6 | 9.9 | 10.9 | **+1.0** | 9.6 | 10.9 | **+1.3** | 9.5 | 10.9 | **+1.4** |
| DS$_{100}$@1.0 | 13.7 | 2.7 | 2.2 | 2.6 | **+0.4** | 2.2 | 2.6 | **+0.4** | 2.1 | 2.6 | **+0.5** |
| DiffPure | 77.5 | 22.4 | 15.7 | 22.1 | **+6.4** | 12.8 | 22.1 | **+9.3** | 11.7 | 22.1 | **+10.4** |

Table S3: Robust accuracy (%) of ResNet-50 on CIFAR-10 under $l_2$-bounded attacks ($\epsilon = 0.5$) for various randomized defenses. Notations are consistent with Table 1.

| Defense | Clean | EoT | $n = 10$ | | | $n = 100$ | | | $n = 1000$ | | |
|---|---|---|---|---|---|---|---|---|---|---|---|
| | | | Rand | Rot | Diff | Rand | Rot | Diff | Rand | Rot | Diff |
| RN@0.25 | 82.5 | 62.7 | 37.5 | 61.0 | **+23.5** | 23.5 | 60.1 | **+36.6** | 15.6 | 58.0 | **+42.4** |
| RN@0.5 | 69.6 | 56.0 | 25.9 | 50.6 | **+24.7** | 11.5 | 48.8 | **+37.3** | 5.5 | 46.5 | **+41.0** |
| RN@1.0 | 51.4 | 43.6 | 11.5 | 33.7 | **+22.2** | 2.8 | 31.9 | **+29.1** | 0.6 | 30.3 | **+29.7** |
| RS$_{100}$@0.25 | 87.0 | 64.2 | 60.9 | 64.4 | **+3.5** | 58.8 | 64.4 | **+5.6** | 57.1 | 64.4 | **+7.3** |
| RS$_{100}$@0.5 | 76.4 | 58.2 | 54.0 | 58.6 | **+4.6** | 51.3 | 58.6 | **+7.3** | 49.4 | 58.6 | **+9.2** |
| RS$_{100}$@1.0 | 60.8 | 42.4 | 37.0 | 41.6 | **+4.6** | 34.2 | 41.6 | **+7.4** | 32.0 | 41.6 | **+9.6** |
| DS$_{100}$@0.25 | 83.3 | 36.4 | 32.2 | 36.2 | **+4.0** | 30.6 | 36.2 | **+5.6** | 29.3 | 36.2 | **+6.9** |
| DS$_{100}$@0.5 | 51.7 | 17.2 | 13.7 | 17.2 | **+3.5** | 12.6 | 17.2 | **+4.6** | 11.9 | 17.2 | **+5.3** |
| DS$_{100}$@1.0 | 12.4 | 8.0 | 7.5 | 7.9 | **+0.4** | 7.3 | 7.9 | **+0.6** | 7.2 | 7.9 | **+0.7** |
| DiffPure | 91.5 | 72.7 | 55.9 | 72.7 | **+16.8** | 48.7 | 72.4 | **+23.7** | 43.9 | 71.3 | **+27.4** |

Table S4: Robust accuracy (%) of ResNet-101 on CIFAR-10 under $l_\infty$-bounded attacks ($\epsilon = \frac{8}{255}$) for various randomized defenses. Notations are consistent with Table 1.

| Defense | Clean | EoT | $n = 10$ | | | $n = 100$ | | | $n = 1000$ | | |
|---|---|---|---|---|---|---|---|---|---|---|---|
| | | | Rand | Rot | Diff | Rand | Rot | Diff | Rand | Rot | Diff |
| RN@0.25 | 83.0 | 33.7 | 15.8 | 33.2 | **+17.4** | 10.4 | 32.8 | **+22.4** | 7.9 | 29.9 | **+22.0** |
| RN@0.5 | 70.2 | 37.3 | 12.1 | 35.1 | **+23.0** | 4.6 | 32.1 | **+27.5** | 2.1 | 26.6 | **+24.5** |
| RN@1.0 | 51.1 | 32.1 | 6.2 | 26.4 | **+20.2** | 1.3 | 21.9 | **+20.6** | 0.3 | 17.9 | **+17.6** |
| $\text{RS}_{100}$@0.25 | 87.2 | 29.7 | 26.9 | 29.5 | **+2.6** | 25.3 | 29.5 | **+4.2** | 24.2 | 29.5 | **+5.3** |
| $\text{RS}_{100}$@0.5 | 77.3 | 36.4 | 32.4 | 36.5 | **+4.1** | 30.0 | 36.5 | **+6.5** | 28.1 | 36.5 | **+8.4** |
| $\text{RS}_{100}$@1.0 | 61.2 | 35.2 | 30.0 | 35.5 | **+5.5** | 26.6 | 35.5 | **+8.9** | 24.1 | 35.5 | **+11.4** |
| $\text{DS}_{100}$@0.25 | 83.6 | 17.6 | 14.9 | 17.6 | **+2.7** | 13.7 | 17.6 | **+3.9** | 12.6 | 17.6 | **+5.0** |
| $\text{DS}_{100}$@0.5 | 52.3 | 21.1 | 18.0 | 20.9 | **+2.9** | 15.8 | 20.9 | **+5.1** | 14.0 | 20.9 | **+6.9** |
| $\text{DS}_{100}$@1.0 | 11.3 | 10.4 | 10.2 | 10.4 | **+0.2** | 10.1 | 10.4 | **+0.3** | 10.0 | 10.4 | **+0.4** |
| DiffPure | 91.7 | 48.6 | 37.0 | 48.2 | **+11.2** | 33.7 | 48.0 | **+14.3** | 31.5 | 45.6 | **+14.1** |

Table S5: Robust accuracy (%) of ResNet-101 on CIFAR-10 under $l_2$-bounded attacks ($\epsilon = 0.5$) for various randomized defenses. Notations are consistent with Table 1.

| Defense | Clean | EoT | $n = 10$ | | | $n = 100$ | | | $n = 1000$ | | |
|---|---|---|---|---|---|---|---|---|---|---|---|
| | | | Rand | Rot | Diff | Rand | Rot | Diff | Rand | Rot | Diff |
| RN@0.25 | 83.0 | 63.6 | 39.4 | 62.8 | **+23.4** | 25.8 | 62.1 | **+36.3** | 18.5 | 60.2 | **+41.7** |
| RN@0.5 | 70.2 | 57.6 | 26.6 | 51.3 | **+24.7** | 12.1 | 49.7 | **+37.6** | 5.8 | 47.5 | **+41.7** |
| RN@1.0 | 51.1 | 44.0 | 11.5 | 34.2 | **+22.7** | 2.6 | 32.3 | **+29.7** | 0.5 | 30.7 | **+30.2** |
| $\text{RS}_{100}$@0.25 | 87.2 | 65.7 | 62.7 | 65.9 | **+3.2** | 60.4 | 65.9 | **+5.5** | 58.8 | 65.9 | **+7.1** |
| $\text{RS}_{100}$@0.5 | 77.5 | 59.4 | 55.3 | 59.5 | **+4.2** | 53.0 | 59.5 | **+6.5** | 50.7 | 59.5 | **+8.8** |
| $\text{RS}_{100}$@1.0 | 61.4 | 42.2 | 37.3 | 42.2 | **+4.9** | 34.8 | 42.2 | **+7.4** | 32.8 | 42.1 | **+9.3** |
| $\text{DS}_{100}$@0.25 | 83.3 | 37.8 | 34.0 | 37.7 | **+3.7** | 32.2 | 37.7 | **+5.5** | 30.6 | 37.7 | **+7.1** |
| $\text{DS}_{100}$@0.5 | 51.8 | 18.7 | 15.3 | 18.8 | **+3.5** | 14.1 | 18.8 | **+4.7** | 13.2 | 18.8 | **+5.6** |
| $\text{DS}_{100}$@1.0 | 11.2 | 9.9 | 9.8 | 10.0 | **+0.2** | 9.7 | 10.0 | **+0.3** | 9.6 | 10.0 | **+0.4** |
| DiffPure | 91.7 | 74.0 | 57.0 | 73.4 | **+16.4** | 49.7 | 73.2 | **+23.5** | 44.8 | 72.1 | **+27.3** |

Table S6: Robust accuracy (%) of Swin-L on ImageNet-1k under $l_\infty$-bounded attacks ($\epsilon = \frac{4}{255}$) for various randomized defenses. Notations are consistent with Table 1.

| Defense | Clean | EoT | $n = 10$ | | | $n = 100$ | | | $n = 1000$ | | |
|---|---|---|---|---|---|---|---|---|---|---|---|
| | | | Rand | Rot | Diff | Rand | Rot | Diff | Rand | Rot | Diff |
| RN@0.25 | 76.2 | 18.5 | 14.5 | 17.6 | **+3.1** | 13.1 | 17.6 | **+4.5** | 12.1 | 17.6 | **+5.5** |
| RN@0.5 | 71.0 | 30.4 | 20.6 | 29.0 | **+8.4** | 17.5 | 28.9 | **+11.4** | 15.4 | 28.7 | **+13.3** |
| RN@1.0 | 59.0 | 33.3 | 20.7 | 31.6 | **+10.9** | 15.6 | 30.8 | **+15.2** | 12.4 | 28.7 | **+16.3** |
| $\text{RS}_{100}$@0.25 | 77.3 | 16.0 | 15.4 | 15.7 | **+0.3** | 15.1 | 15.7 | **+0.6** | 14.9 | 15.7 | **+0.8** |
| $\text{RS}_{100}$@0.5 | 72.5 | 26.6 | 25.2 | 26.4 | **+1.2** | 24.7 | 26.4 | **+1.7** | 24.2 | 26.4 | **+2.2** |
| $\text{RS}_{100}$@1.0 | 63.5 | 31.4 | 29.8 | 31.4 | **+1.6** | 28.8 | 31.4 | **+2.6** | 27.8 | 31.4 | **+3.6** |
| $\text{DS}_{100}$@0.25 | 67.2 | 11.3 | 11.0 | 11.2 | **+0.2** | 10.8 | 11.2 | **+0.4** | 10.7 | 11.2 | **+0.5** |
| $\text{DS}_{100}$@0.5 | 58.2 | 18.7 | 17.4 | 18.9 | **+1.5** | 16.9 | 18.9 | **+2.0** | 16.2 | 18.9 | **+2.7** |
| $\text{DS}_{100}$@1.0 | 32.8 | 15.9 | 14.8 | 16.2 | **+1.4** | 14.4 | 16.2 | **+1.8** | 13.7 | 16.2 | **+2.5** |
| DiffPure | 83.2 | 13.5 | 10.0 | 12.7 | **+2.7** | 9.0 | 12.7 | **+3.7** | 8.2 | 12.7 | **+4.5** |

Table S7: Robust accuracy (%) of Swin-L on ImageNet-1k under $l_2$-bounded attacks ($\epsilon = 2$) for various randomized defenses. Notations are consistent with Table 1.

| Defense | Clean | EoT | $n = 10$ | | | $n = 100$ | | | $n = 1000$ | | |
|---|---|---|---|---|---|---|---|---|---|---|---|
| | | | Rand | Rot | Diff | Rand | Rot | Diff | Rand | Rot | Diff |
| RN@0.25 | 76.5 | 40.7 | 33.2 | 40.9 | **+7.7** | 29.6 | 40.9 | **+11.3** | 27.0 | 40.9 | **+13.9** |
| RN@0.5 | 69.8 | 47.2 | 36.1 | 47.4 | **+11.3** | 30.7 | 47.2 | **+16.5** | 26.8 | 47.0 | **+20.2** |
| RN@1.0 | 59.0 | 45.0 | 29.1 | 44.1 | **+15.0** | 22.0 | 43.4 | **+21.4** | 17.4 | 42.1 | **+24.7** |
| $RS_{100}$@0.25 | 77.2 | 38.3 | 37.5 | 37.8 | **+0.3** | 36.9 | 37.8 | **+0.9** | 36.4 | 37.8 | **+1.4** |
| $RS_{100}$@0.5 | 72.2 | 46.6 | 44.5 | 46.3 | **+1.8** | 43.2 | 46.3 | **+3.1** | 42.4 | 46.3 | **+3.9** |
| $RS_{100}$@1.0 | 62.9 | 43.9 | 41.4 | 43.3 | **+1.9** | 40.1 | 43.3 | **+3.2** | 39.1 | 43.3 | **+4.2** |
| $DS_{100}$@0.25 | 67.6 | 24.3 | 23.4 | 24.1 | **+0.7** | 22.9 | 24.1 | **+1.2** | 22.6 | 24.1 | **+1.5** |
| $DS_{100}$@0.5 | 58.3 | 3.0 | 3.0 | 3.0 | +0.0 | 3.0 | 3.0 | +0.0 | 3.0 | 3.0 | +0.0 |
| $DS_{100}$@1.0 | 32.8 | 6.9 | 6.0 | 6.8 | **+0.8** | 6.0 | 6.8 | **+0.8** | 5.8 | 6.8 | **+1.0** |
| DiffPure | 83.2 | 35.1 | 29.7 | 34.9 | **+5.2** | 27.2 | 34.9 | **+7.7** | 25.8 | 34.9 | **+9.1** |

those under the $l_\infty$ threat model. To test the robustness of our conclusions with respect to model size, we conducted additional experiments using ResNet-101. As shown in Table S4 and Table S5, results under both $l_\infty$ and $l_2$ perturbations remained consistent. We further evaluated the Swin-L, a transformer-based classifier, on ImageNet-1k under $l_\infty$ (Table S6) and $l_2$ (Table S7) perturbation bounds. Across all tested settings, seed-rotation consistently improved robustness over standard randomized deployment.

## C    EXTENSION OF THEORETICAL RESULTS

We now demonstrate that our theoretical results are not restricted to classifiers or adversarial risk. With a generalization of the prediction and evaluation mechanism, our framework extends naturally to any model equipped with an evaluator function and natural risk. Proofs to the theorems in this section are postponed to Section D.

### C.1    EXTENSION BEYOND CLASSIFIERS

We begin by defining a broader notion of a model with evaluator, as an extension to classification model with accuracy metric.

**Definition C.1** (Model with evaluator). Let $h : \mathcal{X} \to \mathcal{Z}$ be a *model*, and $g : \mathcal{Z} \times \mathcal{Y} \to \{0, 1\}$ be an *evaluator*. Then we refer to $h$ as a *model with evaluator*, and the corresponding 0-1 loss $l^{0\text{-}1}((x, y), h) = g(h(x), y)$ is defined for $(x, y) \in \mathcal{X} \times \mathcal{Y}$.

With this definition, the classification model is a model with evaluator where $\mathcal{Z} = \mathcal{Y}$ and the evaluator function is $g(z, y) = \mathbb{1}\{z \neq y\}$.

In Definition C.1, the $l^{0\text{-}1}$ is viewed as an accuracy metric on the model. Then the 0-1 loss of a probabilistic model $\mathbf{h} : \mathcal{X} \to \mathcal{P}(\mathcal{Z})$ is defined in the same way as in Eq. (2):

$$l^{0\text{-}1}((x, y), \mathbf{h}) = \mathbb{E}_{z \sim \mathbf{h}(x)}[g(z, y)]. \tag{20}$$

This generalization enables the definition of risks in the same way as for classifiers. By replacing the output space $\mathcal{Y}$ with a general prediction space $\mathcal{Z}$, and substituting the classification-specific loss with a general evaluator function $g$, all previous results apply to this more general setting.

### C.2    EXTENSION TO NATURAL RISK

We now show that our theoretical results extend beyond adversarial robustness. Specifically, by setting the perturbation set $B(x) = \{x\}$ for all $x \in \mathcal{X}$, we reduce the problem to *natural risk*, *i.e.*, standard model evaluation. The natural risks are defined as follows.

$$\mathcal{R}(\mathbf{h}) = \mathbb{E}_{(x,y) \sim \rho}\big[l^{0\text{-}1}((x, y), \mathbf{h})\big], \quad \mathcal{R}(h) = \mathbb{E}_{(x,y) \sim \rho}\big[l^{0\text{-}1}((x, y), h)\big]. \tag{21}$$

The definitions of vulnerability as in Definition 4.2 are extended to:

$$\mathcal{H}_{vb}(x,y) = \{h \in \mathcal{H}_b : g(h(x),y) = 1\}, \tag{22}$$

$$\mathfrak{H}_{svb}(x,y) = \{\mathcal{H}' \subseteq \mathcal{H}_b : \forall h \in \mathcal{H}', g(h(x),y) = 1\}, \tag{23}$$

$$\mu^{\max}(x,y) = \sup_{\mathcal{H}' \in \mathfrak{H}_{svb}(x,y)} \mu(\mathcal{H}'). \tag{24}$$

In the natural risk setting, the technical condition introduced in Definition 4.3 can be simplified because the following holds.

**Theorem C.1.** *Consider the natural risk $\mathcal{R}$, all models with evaluator are non-degenerating.*

In the natural risk setting, since the only valid adversarial example is the original example $x$ itself, any query attack $Q$ can only return the original input:

$$Q(x,y,h,n) = x, \ \forall h \in \mathcal{H}_b, n \in \mathbb{N}^*. \tag{25}$$

Therefore, the estimated risk under any query attack remains exactly the same as the natural risk:

$$\mathcal{R}(\mathbf{h}_\mu) = \mathbb{E}_{h \sim \mu}\Big[\mathcal{R}_Q^{(n)}(h)\Big], \ \text{ for all valid } Q \text{ and } n \in \mathbb{N}^*. \tag{26}$$

This implies that query attacks have no chance of increasing risk in the natural setting, in contrast to the adversarial case. Theorem 5.1 still holds, but now we can refer to Theorem 4.1 and get a stronger result.

**Theorem C.2.** *Let $\mathbf{h}_\mu$ be a probabilistic model with evaluator on BHS $\mathcal{H}_b$, evaluated under natural risk $\mathcal{R}$. Let $Q$ be a query attack. If $\mathbf{h}_\mu$ satisfies $\mathcal{R}(\mathbf{h}_\mu) < \inf_{h \in \mathcal{H}_b} \mathcal{R}(h)$, then for all $n \in \mathbb{N}^*$ with $n > 1$,*

$$\mathcal{R}^{(n)}(\mathbf{h}_\mu) > \mathcal{R}(\mathbf{h}_\mu) = \mathbb{E}_{h \sim \mu}\Big[\mathcal{R}_Q^{(n)}(h)\Big]. \tag{27}$$

The theorem shows that randomized models incur additional risk from nagging attack, which still holds for natural risk setting, but seed-rotation models remain unaffected.

# D PROOFS

We provide the following results that underpin the main theoretical contributions of this paper.

**Lemma D.1.** *For any probabilistic classifier $\mathbf{h}_\mu$ constructed from $\mathcal{H}_b$ using $\mu \in \mathcal{P}(\mathcal{H}_b)$, if a data point $(x,y) \in (\mathcal{X} \times \mathcal{Y})$ satisfies $\mu^{\max}(x,y) = 1$, then $\pi_{\mathbf{h}_\mu}(x,y) = 0$.*

*Proof.* Recall the definition

$$\pi_{\mathbf{h}_\mu}(x,y) = \mu(\mathcal{H}_{vb}(x,y)) - \mu^{\max}(x,y), \tag{28}$$

Since both terms are probabilities, and $\mu^{\max}(x,y) = 1$, it follows that:

$$\pi_{\mathbf{h}_\mu}(x,y) \le 1 - 1 = 0. \tag{29}$$

On the other hand, the probability of vulnerable set $\mu(\mathcal{H}_{vb}(x,y))$ is always greater than or equal to the probability of commonly vulnerable set $\mu^{\max}(x,y)$, which yields $\pi_{\mathbf{h}_\mu}(x,y) \ge 0$. Therefore, $\pi_{\mathbf{h}_\mu}(x,y) = 0$. $\qquad\square$

**Lemma D.2.** *Let $\mathbf{h}_\mu$ be a non-degenerating probabilistic classifier constructed from $\mathcal{H}_b$ using $\mu \in \mathcal{P}(\mathcal{H}_b)$. Then, for $(x,y) \sim \rho$, we have*

$$\mu^{\max}(x,y) = 0 \Rightarrow \pi_{\mathbf{h}_\mu}(x,y) = 0 \tag{30}$$

*almost surely.*

*Proof.* This follows directly from the converse-negative proposition of the non-degeneracy definition in Definition 4.3. $\qquad\square$

**Lemma D.3.** *Let $\mathbf{h}_\mu$ be a probabilistic classifier over BHS $\mathcal{H}_b$. If $\mathcal{R}_\epsilon(\mathbf{h}_\mu) < \inf_{h \in \mathcal{H}_b} \mathcal{R}_\epsilon(h)$, then for all $n \in \mathbb{N}^*$ with $n > 1$,*

$$\mathcal{R}_\epsilon^{(n)}(\mathbf{h}_\mu) > \mathcal{R}_\epsilon^{(n-1)}(\mathbf{h}_\mu). \tag{31}$$

*Proof.* From Corollary 3.1 of (Gnecco Heredia et al., 2023), a probabilistic classifier satisfies $\mathcal{R}_\epsilon(\mathbf{h}_\mu) < \inf_{h \in \mathcal{H}_b} \mathcal{R}_\epsilon(h)$ if and only if

$$\mathbb{E}_{(x,y) \sim \rho}\left[\pi_{\mathbf{h}_\mu}(x,y)\right] > \mathbb{E}_{h \sim \mu}[\mathcal{R}_\epsilon(h)] - \inf_{h \in \mathcal{H}_b} \mathcal{R}_\epsilon(h). \tag{32}$$

The right-hand side is non-negative, so we conclude that the expected matching penny gap is strictly positive: $\mathbb{E}_{(x,y) \sim \rho}\left[\pi_{\mathbf{h}_\mu}(x,y)\right] > 0$. Now consider the following subtraction when $n > 1$:

$$\mathcal{R}_\epsilon^{(n)}(\mathbf{h}_\mu) - \mathcal{R}_\epsilon^{(n-1)}(\mathbf{h}_\mu) = \mathbb{E}_{(x,y) \sim \rho}\left[\mu^{\max}(x,y)(1 - \mu^{\max}(x,y))^{n-1}\right] \geq 0, \tag{33}$$

since $0 \leq \mu^{\max}(x,y) \leq 1$.

We then prove by contradiction. Suppose $\mathcal{R}_\epsilon^{(n)}(\mathbf{h}_\mu) = \mathcal{R}_\epsilon^{(n-1)}(\mathbf{h}_\mu)$. Then the above expectation must be zero, which implies that $\mu^{\max}(x,y) \in \{0,1\}$ almost surely. Applying Lemma D.1 and Lemma D.2, it follows that $\pi_{\mathbf{h}_\mu}(x,y) = 0$ almost surely, contradicting the earlier conclusion that $\mathbb{E}_{(x,y) \sim \rho}\left[\pi_{\mathbf{h}_\mu}(x,y)\right] > 0$.

Therefore, the assumption must be false, and we conclude:

$$\mathcal{R}_\epsilon^{(n)}(\mathbf{h}_\mu) > \mathcal{R}_\epsilon^{(n-1)}(\mathbf{h}_\mu), \; \forall n > 1. \tag{34}$$

$\square$

**Theorem D.1** (Restatement of Theorem 4.1). *Let $\mathbf{h}_\mu$ be a probabilistic classifier over BHS $\mathcal{H}_b$. If $\mathcal{R}_\epsilon(\mathbf{h}_\mu) < \inf_{h \in \mathcal{H}_b} \mathcal{R}_\epsilon(h)$, then for all $n \in \mathbb{N}^*$ with $n > 1$,*

$$\mathcal{R}_\epsilon^{(n)}(\mathbf{h}_\mu) > \mathcal{R}_\epsilon(\mathbf{h}_\mu). \tag{35}$$

*Proof.* This follows immediately from Lemma D.3. $\square$

Intuitively, the ANR increases monotonically as the query count $n$ grows. This intuition is confirmed by the following lemma.

**Lemma D.4.** *Let $\mathbf{h}_\mu$ be a probabilistic classifier. Then for all $n \in \mathbb{N}^*$ with $n > 1$,*

$$\mathcal{R}_\epsilon^{(n-1)}(\mathbf{h}_\mu) \leq \mathcal{R}_\epsilon^{(n)}(\mathbf{h}_\mu). \tag{36}$$

*Proof.* For all $n \in \mathbb{N}^*, n > 1$, and $(x,y) \in (\mathcal{X} \times \mathcal{Y})$, since the probability $0 \leq \mathbf{h}(x)_y \leq 1$,

$$1 - [\mathbf{h}(x)_y]^{n-1} \leq 1 - [\mathbf{h}(x)_y]^n. \tag{37}$$

So,

$$\sup_{x' \in B_\epsilon(x)} \left\{1 - [\mathbf{h}(x')_y]^{n-1}\right\} \leq \sup_{x' \in B_\epsilon(x)} \left\{1 - [\mathbf{h}(x')_y]^n\right\}. \tag{38}$$

Taking expectation $\mathbb{E}_{(x,y) \sim \rho}$ on both sides completes the proof. $\square$

**Theorem D.2** (Restatement of Theorem 4.2). *Let $\mathbf{h}_\mu$ be a probabilistic classifier. Then for all $n \in \mathbb{N}^*$,*

$$\mathcal{R}_\epsilon(\mathbf{h}_\mu) \leq \mathcal{R}_\epsilon^{(n)}(\mathbf{h}_\mu) \leq 1 - (1 - \mathcal{R}_\epsilon(\mathbf{h}_\mu))^n. \tag{39}$$

*Proof.* The left inequality follows from repeated application of Lemma D.4:

$$\mathcal{R}_\epsilon(\mathbf{h}_\mu) = \mathcal{R}_\epsilon^{(1)}(\mathbf{h}_\mu) \leq \mathcal{R}_\epsilon^{(2)}(\mathbf{h}_\mu) \leq \cdots \leq \mathcal{R}_\epsilon^{(n)}(\mathbf{h}_\mu). \tag{40}$$

For the right inequality, using Jensen's inequality:

$$\mathcal{R}_\epsilon^{(n)}(\mathbf{h}_\mu) = \mathbb{E}_{(x,y)\sim\rho}\left(\sup_{x'\in B_\epsilon(x)}\left(1 - [\mathbf{h}(x')_y]^n\right)\right) \tag{41}$$

$$= 1 - \mathbb{E}_{(x,y)\sim\rho}\left(\inf_{x'\in B_\epsilon(x)}[\mathbf{h}(x')_y]^n\right) \tag{42}$$

$$= 1 - \mathbb{E}_{(x,y)\sim\rho}\left(\inf_{x'\in B_\epsilon(x)}\mathbf{h}(x')_y\right)^n \tag{43}$$

$$= 1 - \mathbb{E}_{(x,y)\sim\rho}\left(1 - \sup_{x'\in B_\epsilon(x)}(1 - \mathbf{h}(x')_y)\right)^n \tag{44}$$

$$\leq 1 - \left(\mathbb{E}_{(x,y)\sim\rho}\left(1 - \sup_{x'\in B_\epsilon(x)}(1 - \mathbf{h}(x')_y)\right)\right)^n \tag{45}$$

$$= 1 - \left(1 - \mathbb{E}_{(x,y)\sim\rho}\left(\sup_{x'\in B_\epsilon(x)}(1 - \mathbf{h}(x')_y)\right)\right)^n \tag{46}$$

$$= 1 - (1 - \mathcal{R}_\epsilon(\mathbf{h}_\mu))^n, \tag{47}$$

since $f(x) = x^n$ is convex on $\mathbb{R}$ when $n \geq 1$. $\qquad\square$

**Lemma D.5.** *Let $\mathbf{h}_\mu$ be a probabilistic classifier. Then,*

$$\lim_{n\to+\infty}\mathcal{R}_\epsilon^{(n)}(\mathbf{h}_\mu) = \mathbb{P}_{(x,y)\sim\rho}(\mu^{\max}(x,y) > 0). \tag{48}$$

*Proof.* From the proof to Theorem 3.2 of (Gnecco Heredia et al., 2023) we know that:

$$\sup_{x'\in B_\epsilon(x)}\mathbb{E}_{h\sim\mu}[\mathbb{1}\{h(x') \neq y\}] = \sup_{x'\in B_\epsilon(x)}(1 - \mathbf{h}(x')_y) = \mu^{\max}(x,y). \tag{49}$$

Then the nagging risk is rewritten as:

$$\mathcal{R}_\epsilon^{(n)}(\mathbf{h}_\mu) = \mathbb{E}_{(x,y)\sim\rho}\left[1 - \left(1 - \sup_{x'\in B_\epsilon(x)}(1 - \mathbf{h}(x')_y)\right)^n\right] \tag{50}$$

$$= \mathbb{E}_{(x,y)\sim\rho}[1 - (1 - \mu^{\max}(x,y))^n]. \tag{51}$$

Since $0 \leq \mu^{\max}(x,y) \leq 1$,

$$\lim_{n\to+\infty}(1 - \mu^{\max}(x,y))^n = \begin{cases} 1 & \text{if } \mu^{\max}(x,y) = 0, \\ 0 & \text{if } \mu^{\max}(x,y) > 0. \end{cases} \tag{52}$$

By the dominated convergence theorem, since $0 \leq 1 - (1 - \mu^{\max}(x,y))^n \leq 1, \forall n \in \mathbb{N}^*$, we can exchange the limit and expectation:

$$\lim_{n\to+\infty}\mathcal{R}_\epsilon^{(n)}(\mathbf{h}_\mu) = \lim_{n\to+\infty}\mathbb{E}_{(x,y)\sim\rho}[1 - (1 - \mu^{\max}(x,y))^n] \tag{53}$$

$$= \mathbb{E}_{(x,y)\sim\rho}\left[1 - \lim_{n\to+\infty}(1 - \mu^{\max}(x,y))^n\right] \tag{54}$$

$$= \mathbb{E}_{(x,y)\sim\rho}[\mathbb{1}\{\mu^{\max}(x,y) > 0\}] \tag{55}$$

$$= \mathbb{P}_{(x,y)\sim\rho}[\mu^{\max}(x,y) > 0]. \tag{56}$$

$\qquad\square$

**Theorem D.3** (Restatement of Theorem 4.3). *Let $\mathbf{h}_\mu$ be a non-degenerating probabilistic classifier. Then,*

$$\lim_{n\to+\infty}\mathcal{R}_\epsilon^{(n)}(\mathbf{h}_\mu) = \mathbb{P}_{(x,y)\sim\rho}(\exists x' \in B_\epsilon(x), h \in \mathcal{H}_b \text{ such that } h(x') \neq y). \tag{57}$$

*Proof.* The non-degeneracy condition ensures the equivalence

$$\mu[\mathcal{H}_{vb}(x,y)] > 0 \Leftrightarrow \mu^{\max}(x,y) > 0 \tag{58}$$

almost surely over the data distribution $\rho$. The forward implication is guaranteed by the assumption of non-degeneracy. The reverse implication always holds because

$$\mu[\mathcal{H}_{vb}(x,y)] \geq \mu^{\max}(x,y) \tag{59}$$

by definition. $\square$

**Theorem D.4** (Restatement of Theorem 4.4). *Let $\mathbf{h}_\mu$ be an REC on $M$ base hypotheses $h_1, h_2, \ldots, h_M$ with uniform prior $\mu(h_i) = \frac{1}{M}$, $i = 1, 2, \ldots, M$. Let $(x, y)$ be the data point that we focus on. Assume for each base hypothesis $h_i$, there exists an disjoint-vulnerable adversarial example $x'_i$, s.t., $h_i(x'_i) \neq y, h_j(x'_i) = y$, $\forall j \neq i$, i.e., the adversarial example is specific to that hypothesis. Assume the pairwise Kullback–Leibler divergence is uniformly bounded:*

$$D_{KL}(P_{i,x'} \| P_{j,x'}) \leq d_{\max} < \infty, \ \forall i \neq j, x' \in B_\epsilon(x), \tag{60}$$

*where $P_{i,x'}$ is the logit of model $h_i$ with regard to input $x'$. Then, for the specific data point $(x, y)$,*

$$\mathcal{R}_\epsilon^{(n)}(\mathbf{h}_\mu) \leq \frac{n d_{\max} + \log 2}{\log M}. \tag{61}$$

*Equivalently, to guarantee $\mathcal{R}_\epsilon^{(n)}(\mathbf{h}_\mu) \geq 1 - \delta$ one must have*

$$n \geq \frac{(1 - \delta) \log M - \log 2}{d_{\max}}. \tag{62}$$

*Proof.* The inequality is achieved by Faro's inequality in information theory. To prove the theorem, we first link the success of attack to the identification of hypothesis $h_i$, then bound the ANR by mutual information, and finally bound the mutual information by KL divergence.

First of all, based on the disjoint-vulnerability assumption, the event of at least one success implies the event of correct identification of the hypothesis $h_i$. Thus, ANR $\mathcal{R}_\epsilon^{(n)}(\mathbf{h}_\mu)$ is upper-bounded by the probability of correct identification that any $n$-query strategy can achieve. So it suffices to bound the probability of correct identification among the $M$ possible hypotheses after $n$ queries. This let us use Fano's inequality.

The next step is to apply Fano's inequality for an information-theoretic bound. Let $\hat{h}$ be an estimator of $h \in \{h_1, h_2, \ldots, h_M\}$ based on the $n$-nagging transcript $Y^n$. Denote the probability of error $P_e = \mathbb{P}(\hat{h} \neq h)$. Fano's inequality gives:

$$H(h|Y^n) \leq H_b(P_e) + P_e \log(M - 1) \leq \log 2 + P_e \log M, \tag{63}$$

where $H$ is the entropy and $H_b$ the binary entropy. Rearranging and using uniform prior $H(h) = \log M$ yields:

$$\log M - I(h; Y^n) \leq \log 2 + P_e \log M, \tag{64}$$

i.e.

$$\mathbb{P}(\hat{h} = h) \leq \frac{I(h; Y^n) + \log 2}{\log M}. \tag{65}$$

Since ANR is upper-bounded by the probability of correct identification,

$$\mathcal{R}_\epsilon^{(n)}(\mathbf{h}_\mu) \leq \frac{I(h; Y^n) + \log 2}{\log M}. \tag{66}$$

Then we need to give a bound on the mutual information $I(h; Y^n)$. Using the mutual information chain rule:

$$I(h; Y^n) = \sum_{t=1}^n I(h; Y_t | Y^{t-1}), \tag{67}$$

where $Y_t$ is the transcript of $t$-th nagging query. Given the previous transcript $Y^{t-1}$, the current distribution of logit should be a mixture of logits based on the conditional distribution of hypotheses $h$, and recall that logit is denoted as $P_{i,x'}$. Let

$$\bar{P}_t = \mathbb{P}(Y_t|Y^{t-1}) = \sum_i \mathbb{P}(h = h_i|Y^{t-1})P_{i,x'}. \tag{68}$$

By definition of conditional mutual information,

$$I(h; Y_t|Y^{t-1}) = \mathbb{E}_{Y^{t-1}}\left[\sum_i \mathbb{P}(h = h_i|Y^{t-1})D_{\mathrm{KL}}(\mathbb{P}(Y_t|h = h_i, Y^{t-1})\|\bar{P}_t)\right]. \tag{69}$$

Using the convexity of KL divergence, we have

$$I(h; Y_t|Y^{t-1}) \leq \mathbb{E}_{Y^{t-1}}\left[\sum_{i,j} \mathbb{P}(h = h_i|Y^{t-1})\mathbb{P}(h = h_j|Y^{t-1})D_{\mathrm{KL}}(P_{i,x'}\|P_{j,x'})\right]. \tag{70}$$

Since we assume the KL divergence is uniformly bounded by $d_{\max}$, we have

$$I(h; Y_t|Y^{t-1}) \leq d_{\max}. \tag{71}$$

Chaining the $n$ conditional mutual information together yields

$$I(h; Y^n) \leq n d_{\max}. \tag{72}$$

Finally, substitute the bound into Eq. (66):

$$\mathcal{R}_\epsilon^{(n)}(\mathbf{h}_\mu) \leq \frac{n d_{\max} + \log 2}{\log M}. \tag{73}$$

This completes the proof. □

EoT-based query attack $Q_{\mathrm{eot}}(\epsilon)$ initializes by constructing an adversarial input that is "universal" for all deterministic classifier $h \in \mathcal{H}_b$:

$$Q_{\mathrm{eot}}(\epsilon)(x, y, h, 1) \coloneqq \arg\max_{x' \in B_\epsilon(x)} (1 - \mathbf{h}(x')_y), \ \forall h \in \mathcal{H}_b. \tag{74}$$

In practice, this corresponds to constructing an adversarial example using an offline EoT-based white-box attack. Once generated, the adversarial input $x'$ is evaluated against an online black-box model sampled from $\mu$, which results in an estimated adversarial risk

$$\mathbb{E}_{h\sim\mu}\left[\mathcal{R}_{Q_{\mathrm{eot}}(\epsilon)}^{(1)}(h)\right] = \mathcal{R}_\epsilon(\mathbf{h}_\mu). \tag{75}$$

Then we have the following bound on query risk.

**Theorem D.5.** *Let $\mathbf{h}_\mu$ be a probabilistic classifier. Let $Q$ be any monotonic query attack that initializes with $Q_{eot}$. Then for all $n \in \mathbb{N}^*$,*

$$\mathcal{R}_\epsilon(\mathbf{h}_\mu) \leq \mathbb{E}_{h\sim\mu}\left[\mathcal{R}_{Q(\epsilon)}^{(n)}(h)\right] \leq \mathbb{E}_{h\sim\mu}[\mathcal{R}_\epsilon(h)]. \tag{76}$$

*Proof.* The lower bound follows because the initial adversarial example from $Q_{\mathrm{eot}}$ is crafted to maximize misclassification over distribution $h_\mu$, which as a strong initialization for any monotonic query strategy. Therefore, the query-based risk over deterministic $h \sim \mu$ with this initialization is at least as high as the base risk of $\mathbf{h}_\mu$.

The upper bound holds because $\mathcal{R}_\epsilon(h)$ is defined as the existence of adversarial examples within the allowed perturbation, while a black-box attack $Q(\epsilon)$ may not discover all possible adversarial examples. Hence, $\mathcal{R}_{Q(\epsilon)}^{(n)}(h) \leq \mathcal{R}_\epsilon(h)$ for any $h$, and taking the expectation completes the proof. □

**Theorem D.6** (Restatement of Theorem 5.1). *Let $\mathbf{h}_\mu$ be a non-degenerating probabilistic classifier, and let $Q$ be any query attack. Then,*

$$\lim_{n\to+\infty} \mathbb{E}_{h\sim\mu}\left[\mathcal{R}_{Q(\epsilon)}^{(n)}(h)\right] \leq \mathbb{E}_{h\sim\mu}[\mathcal{R}_\epsilon(h)] \leq \lim_{n\to+\infty} \mathcal{R}_\epsilon^{(n)}(\mathbf{h}_\mu). \tag{77}$$

*Proof.* The left inequality follows directly from the definition of $\mathcal{R}_\epsilon(h)$ as the existence of adversarial examples, which is at least as high as that achieved by any specific query attack $Q$, even with unlimited queries.

The right inequality is a consequence of Lemma D.5 and the fact that

$$\mathbb{E}_{h\sim\mu}[\mathcal{R}_\epsilon(h)] = \mathbb{E}_{(x,y)\sim\rho}[\mu[\mathcal{H}_{vb}(x,y)]], \tag{78}$$

which is shown in (Gnecco Heredia et al., 2023). Then, non-degeneracy ensures $\mu^{\max}(x,y) = 0 \Leftrightarrow \mu(\mathcal{H}_{vb}(x,y)) = 0$ almost surely, and the right inequality follows. $\qquad\square$

**Proposition D.1** (Restatement of Proposition A.1). *An REC $\mathbf{h}_\mu$ is always a non-degenerating probabilistic classifier.*

*Proof.* A REC is a finite ensemble of $M$ base classifiers. Suppose a data point $(x,y)$ satisfies $\mu[\mathcal{H}_{vb}(x,y)] > 0$. Then there must exist some $h_0 \in \mathcal{H}_{vb}$ such that $\mu[\{h_0\}] > 0$. Since $\{h_0\} \in \mathfrak{H}_{svb}(x,y)$, it follows that

$$\mu^{\max}(x,y) \geq \mu(\{h_0\}) > 0. \tag{79}$$

Hence, the REC is non-degenerating. $\qquad\square$

**Proposition D.2** (Restatement of Proposition A.2). *A WNC $\mathbf{h}_\mu$ with base mapping $h$ that is continuous on $\mathcal{W} \times \mathcal{X}$ is non-degenerating, provided that the adversarial input $x'$ misclassifies the data point $(x,y)$ in a non-vanishing way. That is, the total predicted probability of the incorrect labels remains strictly higher than the predicted probability of the true label almost surely.*

*Proof.* From the conditions, there exists a data point $(x,y)$ such that $\mu[\mathcal{H}_{vb}(x,y)] > 0$. Then there exists a misclassifying hypothesis $h_0 \in \mathcal{H}_{vb}$ constructed by some weights $w_0 \in \mathcal{W}$. By continuity of $h$, the classifier output varies continuously with $w \in \mathcal{W}$. Given that $x'$ leads to confident misclassification under $h_{w_0}$, there exists a neighborhood

$$\mathcal{H}_D = \{h_w : w \in \mathcal{W}, d(w, w_0) < \delta\} \tag{80}$$

such that $h_w(x') \neq y$ for all $h_w \in \mathcal{H}_D$. Since $\mathcal{H}_D \in \mathfrak{H}_{svb}(x,y)$ and $\mu[\mathcal{H}_D] > 0$, it follows that

$$\mu^{\max}(x,y) \geq \mu[\mathcal{H}_D] > 0. \tag{81}$$

Thus, the WNC is non-degenerating. $\qquad\square$

**Lemma D.6.** *Let $\mathcal{H}_b$ be a BHS. Consider the natural risk $\mathcal{R}$. Then, for all $(x,y) \in \mathcal{X} \times \mathcal{Y}$,*

$$\mu^{\max}(x,y) = \mu(\mathcal{H}_{vb}(x,y)). \tag{82}$$

*Proof.* We begin by showing that every subset $\mathcal{H}' \in \mathfrak{H}_{svb}(x,y)$ satisfies $\mathcal{H}' \subseteq \mathcal{H}_{vb}(x,y)$. This is proven by contradiction. Suppose that there exists some $\mathcal{H}' \in \mathfrak{H}_{svb}(x,y)$ and a hypothesis $h \in \mathcal{H}' \backslash \mathcal{H}_{vb}(x,y)$. By the definiton of $\mathfrak{H}_{svb}(x,y)$, all $h \in \mathcal{H}'$ must satisfy $g(h(x),y) = 1$. However, $h \notin \mathcal{H}_{vb}(x,y)$ implies $g(h(x),y) \neq 1$, which is a contradiction. Hence, the assumption is invalid. Therefore,

$$\mu(\mathcal{H}') \leq \mu(\mathcal{H}_{vb}(x,y)), \ \forall \mathcal{H}' \in \mathfrak{H}_{svb}(x,y), \tag{83}$$

by the definition of probability measure. Thus,

$$\mu^{\max}(x,y) = \sup_{\mathcal{H}' \in \mathfrak{H}_{svb}(x,y)} \mu(\mathcal{H}') \leq \mu(\mathcal{H}_{vb}(x,y)). \tag{84}$$

Now, observe that $\mathcal{H}_{vb}(x,y) \in \mathfrak{H}_{svb}(x,y)$ by definition, so $\mu^{\max}(x,y) \geq \mu(\mathcal{H}_{vb}(x,y))$. Combining both inequalities yields the desired equality $\mu^{\max}(x,y) = \mu(\mathcal{H}_{vb}(x,y))$. $\qquad\square$

**Theorem D.7** (Restatement of Theorem C.1). *Under the natural risk $\mathcal{R}$, all models with evaluator are non-degenerating.*

*Proof.* This follows immediately from Lemma D.6, which shows the condition for non-degeneracy (as defined in Definition 4.3) is satisfied for all $(x,y)$. $\qquad\square$

**Theorem D.8** (Restatement of Theorem C.2). *Let $\mathbf{h}_\mu$ be a probabilistic model with evaluator on BHS $\mathcal{H}_b$, evaluated under natural risk $\mathcal{R}$. Let $Q$ be a query attack. If $\mathbf{h}_\mu$ satisfies $\mathcal{R}(\mathbf{h}_\mu) < \inf_{h \in \mathcal{H}_b} \mathcal{R}(h)$, then for all $n \in \mathbb{N}^*$ with $n > 1$,*

$$\mathcal{R}^{(n)}(\mathbf{h}_\mu) > \mathcal{R}(\mathbf{h}_\mu) = \mathbb{E}_{h \sim \mu}\Big[\mathcal{R}_Q^{(n)}(h)\Big]. \tag{85}$$

*Proof.* The left inequality follows by applying the natural-risk counterpart of Theorem 4.1, replacing adversarial risk with natural risk. The right equality follows because query attacks cannot increase risk in the natural setting, as formalized in Eq. (26). $\qquad\square$