# OpenReview forum: "On the Inherent Vulnerability of Randomized Models to Nagging Attack"
_ICLR.cc/2026/Conference — Submitted to ICLR 2026_

### Official Review · Reviewer_B9t3 · 2025-10-30

**Soundness:** 3
**Presentation:** 3
**Contribution:** 3
**Rating:** 6
**Confidence:** 3

**Summary:**

This paper studies how randomness at inference time can actually make models less safe. It introduces the nagging attack, where an adversary keeps sending the same input until a randomized model fails. The authors formalize this with Adversarial Nagging Risk (ANR) and prove that any nontrivial probabilistic model becomes more vulnerable once multiple queries are allowed.
To mitigate this, they propose seed-rotation—fixing a random seed for short intervals (e.g., per day) and refreshing it periodically. Both theory and experiments show that this approach makes randomized models significantly safer without changing the underlying model.

**Strengths:**

+ Offers a clear theoretical framework showing why randomness increases risk.
+ The seed-rotation idea is simple, practical, and effective.
+ Writing is clear and well-organized, making complex math easy to follow.

**Weaknesses:**

- The rotation frequency is chosen heuristically, with no clear analysis of how interval length, query rate, or system overhead affect robustness or performance.
- The defense’s resilience to adaptive attackers is unclear—an adversary could potentially infer or exploit the fixed seed during each interval.
- The paper lacks comparisons and ablations: it doesn’t evaluate against other mitigations (e.g., query tracking, adaptive noise scaling) or isolate which randomness source drives vulnerability.
- Theoretical claims are strong, but empirical validation and formal guarantees of ANR reduction remain limited.

**Questions:**

1.	How was the rotation frequency chosen? Is there a measurable relationship between interval length, attacker query rate, and resulting ANR?
2.	Does frequent seed-rotation introduce computational or caching overheads in real deployments, and how do these costs compare to the achieved robustness gains?
3.	Could an adaptive adversary infer or approximate the current seed or its distribution from model outputs, potentially weakening the defense?
4.	Seed-rotation makes the model deterministic within intervals—how does this trade off between reduced nagging risk and increased vulnerability to query-based black-box attacks?
5.	The theory predicts exponential degradation with query count. Is this trend empirically verified across all defenses, datasets, and perturbation bounds?
6.	Does seed-rotation offer any formal probabilistic bounds on ANR reduction, or are the observed improvements purely empirical?
7.	Why weren’t other mitigations such as query tracking, adaptive noise scaling, or deterministic ensembles included for comparison? Could these achieve similar robustness without rotation?

---

> ### Author Response · Authors · 2025-11-27
>
> We sincerely thank the reviewer for the valuable and constructive feedback. Below, we respond point-by-point to the concerns raised:
>
> 1.1. **Is there a measurable relationship between interval length, attacker query rate, and resulting ANR?** Yes. As noted by the reviewer, an adversary could potentially infer or exploit the fixed seed during each interval. Let $r$ be the query rate and $l$ the interval length, the adversary's query budget to exploit the fixed seed is $n=lr$. A larger $n$ naturally increases the adversary’s ability to exploit the fixed seed. Importantly, this is not the same as the ANR metric $R_{\epsilon}^{ (n) } (h_\mu)$, which is defined on randomized deployment scheme. For the seed-rotation scheme, we evaluated the query attack against the now-deterministic model, whose query budget is also denoted as $n$ (Tables 1&2). Within 1000 queries, the ANR increased by 3.9% (RN@0.25, on Line 386).
>
> 1.2. **How was the rotation frequency chosen?** We chose the rotation frequency to be 1 day purely for demonstration purpose. The evaluation of ANR did not depend on the actual rotation frequence in deployment. As responded above, there is a measurable relationship between interval length, attacker query rate, and resulting ANR. Thus, the parameter reduces to query budget $n$, which was set to 10,100,1000 in our experiments. The rotation frequency is calculated from the expected query budget and attacker query rate limit in the real system.
>
> 2. **Does frequent seed-rotation introduce computational or caching overheads in real deployments?** The only extra computation is to generate a new seed, typically a 32-bit integer, and use it to reset the state of the RNG. In fact, the seed-rotation scheme is more cache-friendly than the randomized counterpart. The randomized counterpart cannot be cached since the output varies for each query, whereas seed-rotation allows caching within a rotation interval.
>
> 3. **Could an adaptive adversary infer or approximate the current seed or its distribution from model outputs?** This is an insightful question. First, the seed distribution is known to the adversary in our threat model. Second, there is a possibility that the seed may be approximated by probing model outputs, like a cryptanalysis attack. To our best knowledge, no prior work has studied seed-recovery attacks in this setting. We identify this as an interesting direction for future research.
>
> 4. **Trade off between reduced nagging risk and increased vulnerability to query-based black-box attacks**: It depends on the deployment to select proper interval length. We evaluated effective query count of black-box attacks and nagging count both up to $n=1000$. Based on our observation (Tables 1&2), accuracy under nagging attack degrades almost exponentially with $n$; accuracy under query attack degrades much slower, mostly because hard-label query-based black-box attacks often needs a lot of queries to find an adversarial example (Chen&Gu, 2020).
>
> 5. The theory predicts an exponential accuracy lower bound with nagging query count. The actual degradation depends on internal factor like the failure probability of each data point, e.g., some data points are certified to be robust. Empirically, we found that the degradation is slower than an exponential decay (aligned with prediction). This is even more concerning -- most vulnerabilities are exposed within 10 nagging queries, e.g., 18.4% accuracy degradation from $n=1$ (EoT) to $n=10$ (RN@0.25, on Line 386).
>
> 6. **Does seed-rotation offer any formal probabilistic bounds on ANR reduction, or are the observed improvements purely empirical?** The observed improvements are purely empirical. As mentioned above, the additional vulnerability of seed-rotation is query attack against the now-deterministic model, whose accuracy lower bound is the white-box robust accuracy of the model. We did not evaluate white-box since we assume the current seed is kept secret to the adversary. We managed to establish a formal comparison between seed-rotation and randomized deployment in Thm. 5.1, which claims that seed-rotation is ultimately safer than randomization in the limit as $n\to\infty$.
>
> 7. These methods address query-based attacks but are incompatible with our setting. We focus on directly mitigating the vulnerability of randomization. Query tracking avoids attack by detecting abnormal query sequence and restricting use -- it remains evadable with distributed accounts, caching, or query batching. Similarly, adaptive noise scaling may be evaded with distributed accounts. Deterministic ensemble is a deterministic defense that we did not discuss in this paper. Note that we did not compare randomized defenses to deterministic ones. Prevailing deterministic defense methods like adversarial training do not suffer from ANR. Our contribution is that we analyze a fundamental vulnerability of randomized models and provide a new pathway for safely deploying them.

---

### Official Review · Reviewer_2WXW · 2025-10-31

**Soundness:** 2
**Presentation:** 4
**Contribution:** 2
**Rating:** 4
**Confidence:** 4

**Summary:**

The paper analyzes the adversarial risk induced by nagging attacks against randomized models. Both theoretical and empirical justifications are given to show that adversarial risk increases in proportion with the query budget. A seed-rotation strategy is proposed to address the vulnerability caused by randomness at inference time.

**Strengths:**

The proposed study answers very important questions about potential vulnerability as a result of randomness at inference time.
Under specified assumptions, the theoretical justifications are convincing and the empirical results are promising.

**Weaknesses:**

The definition of adversarial nagging risk is not rigorous and needs further justification.
The experimental setup needs to be improved.

**Questions:**

1. The adversarial nagging risk (ANR) is deliberately defined (Equation 6) to amplify risk as the query budget increases, even though the predictions of a classifier remain the same. Consider a randomized classifier that predicts m times for each input query x and then outputs the majority vote with a probability (tampered with a small random noise). Will the query budgets of 10 and 100 make any difference?  Why should the risks be different in these two cases given that the outcome is the same? This definition of ANR needs to be carefully justified since it has cascading impact on the theorems hereafter.

2. In the randomized deployment, a fixed adversarial sample is tested with n random instances of a classification model, while in the seed-rotation deployment, different adversarial samples crafted in n queries within a single interval are tested with a single model. The latter does not seem to be allowed to explore the vulnerability of randomness in the instances of the classification model. It would be interesting to find out what happens if, in the seed-rotation deployment, after x_0' is crafted for h_0,  the classifiers in later intervals h_1, h_2, ..., h_n also predict for  x_0' and count the number of attack successes.

---

> ### Author Response · Authors · 2025-11-27
>
> We are very grateful for your constructive feedback. Below, we respond point-by-point to the concerns raised:
>
> 1. The randomized classifier considered by the reviewer is essentially Randomized Smoothing (RS). In this scheme, all the queries for the majority vote is considered as one logical query. However, because each query draws a fresh set of noise samples, the majority vote itself can vary across queries, and the output of the RS classifier for "query 1" vs. "query 2" may differ. This variability means that ANR remains well-defined, even for RS. Details on the definition of probabilistic classifier are provided in Line 151. As shown in the table below, ANR differs depending on whether the majority vote uses 1, 10, or 100 samples. (Settings are aligned with Table 1.)
>
> | Defense          | Clean | EoT  | Rand (n=10) | Rot (n=10) | Diff (n=10) | Rand (n=100) | Rot (n=100) | Diff (n=100) | Rand (n=1000) | Rot (n=1000) | Diff (n=1000) |
> | ---------------- | ----- | ---- | ----------- | ---------- | ----------- | ------------ | ----------- | ------------ | ------------- | ------------ | ------------- |
> | RS$_{1}$@0.5       | 69.6  | 36.4 | 12.1        | 34.6       | +22.5       | 4.5          | 31.9        | +27.4        | 2.1           | 26.5         | +24.4         |
> | RS$_{10}$@0.5  | 75.5  | 40.0 | 27.2        | 40.0       | +12.8       | 20.6         | 40.0        | +19.4        | 16.5          | 40.0         | +23.5         |
> | RS$_{100}$@0.5 | 76.6  | 35.9 | 31.7        | 35.9       | +4.2        | 29.2         | 35.9        | +6.7         | 27.3          | 35.9         | +8.6          |
>
> 2. Below we list the accuracy of $x_0'$ targeting $h_0$ and evaluated on $h_1, h_2, ..., h_5$ (independently sampled seeds). The accuracy of subsequent models are similar to $h_0$. This is because the initial adversarial examples were already crafted with EoT of models, and they are universal to all random seeds. In contrast, additional adversarial examples by query attack are specific to $h_0$, so an accuracy increase from 31.5 to 34.8 is observed from $h_0$ to $h_1$ on RN@0.5. We will add this experiment to the revised version. (Settings are aligned with Table 1.)
>
> | Defense          | 0 (Original seed) | 1    | 2    | 3    | 4    | 5    | Mean of 1~5 |
> | ---------------- | ------------ | ---- | ---- | ---- | ---- | ---- | ---------------- |
> | RN@0.5     | 31.5         | 34.8 | 34.9 | 34.8 | 34.6 | 35.0 | 34.8             |
> | RS$_{100}$@0.5 | 35.9         | 35.9 | 35.9 | 35.9 | 35.9 | 35.9 | 35.9             |
> | DS$_{100}$@0.5 | 21.2         | 21.2 | 21.2 | 21.1 | 21.2 | 21.1 | 21.2             |
> | Diffpure     | 48.5         | 48.9 | 48.8 | 48.9 | 48.9 | 48.8 | 48.9             |

---

### Official Review · Reviewer_kkTL · 2025-11-01

**Soundness:** 2
**Presentation:** 3
**Contribution:** 2
**Rating:** 4
**Confidence:** 4

**Summary:**

The authors demonstrate that randomized models are susceptible to a “nagging attack,” in which an adversary can make multiple attempts to breach the model. This vulnerability, however, can be reduced through key-protected deployment of randomized models.

**Strengths:**

1. The paper is clearly written and well organized.
2. The notations and definitions are presented in a straightforward and understandable manner.

**Weaknesses:**

Essentially, this paper demonstrates that a probabilistic classifier faces a higher nagging risk (as defined in Definition 4.1) when the number of allowed queries approaches infinity. Implementing key-guarded deployment—where the model is fixed to a specific realization from the BHS for a given time period—reduces the effectiveness of queries in identifying the worst-case model realization, thereby improving performance under the nagging attack threat model. The main issues are:

1. This result is somewhat trivial and aligns with expectations.
2. I don’t think the nagging risk is a suitable metric for evaluating model robustness. In this framework, any model with a non-zero probability of failure will exhibit the same nagging risk as a model that fails on every attempt when the query budget approaches infinity. However, it’s clear that a model with a smaller failure probability is still preferable to one that fails consistently.
3. According to the threat model definition, the adversary possesses full white-box knowledge of the model, including the defense mechanism, except for the random seed. In that case, the adversary could construct a substitute model without the key-guarded deployment, effectively eliminating the “less efficient query” effect and arriving at the same adversarial example as if the key-guarded deployment were not used.

**Questions:**

1. How are seeds rotated in obtaining the experiment results in the paper? How would rotating with different circles affect the performance?

---

> ### Author Response · Authors · 2025-11-27
>
> Thank you for the careful assessment of our work. Below we address the questions.
>
> 1.1. **Essentially, this paper demonstrates that a probabilistic classifier faces a higher nagging risk (as defined in Definition 4.1) when the number of allowed queries approaches infinity.**: We would like to clarify that our results do not rely on the number of allowed queries approaching infinity. The main theorem, Thm. 4.1, holds for any **finite** $n\ge 1$. Only the second property discusses asymptotic behavior as $n$ approaches infinity. All experimental evaluations use finite values of $n$.
>
> 1.2. **This result is somewhat trivial and aligns with expectations**: We agree that the result aligns with intuition -- our contribution is to formalize this intuition, quantify it precisely, and demonstrate that prior work on randomized defenses did not recognize or evaluate this vulnerability. Despite the intuitive nature of the phenomenon, its implications for deployment were previously overlooked. Our work fills this gap and provides a new pathway for safely deploying randomized models with our deployment protocol, termed seed-rotation deployment.
>
> 2.1. **Significance of the nagging risk metric**: One may link the failure probability $R_{\epsilon} (h_\mu)$ with the nagging risk metric $R_{\epsilon}^{ (n) } (h_\mu)$ via the first property (Thm 4.2). Let us compare two randomization methods $h_1$ and $h_2$ under the same $n$. The one with higher failure probability $R_{\epsilon} (h_\mu)$ has higher upper&lower bounds on nagging risk. But the actual value of nagging risk depends on the specific method, and it cannot be inferred solely from single-query failure probability. Our experiments highlight this nontrivial behavior (Table 1):
> - RN@0.25 has a lower failure probability than RS$_{100}$@0.25 (accuracy is 32.4 vs. 28.5).
> - Yet for ANR with $n=10$, the risk of RN is much higher than RS (accuracy 14.0 vs. 25.8).
>
> 2.2. Correctly noted by the reviewer, in some extreme case, *a model with a non-zero probability of failure will exhibit the same nagging risk as a model that fails on every attempt when the query budget approaches infinity*. But our focus is the practical regime of finite query budgets, which matches real-world attacker capabilities. Theory and experiments are mainly designed for this finite-$n$ setting.
>
> 3. Our threat model concerns attacking a remote deployed model rather than a local white-box copy. To ensure a strong and conservative evaluation, we assume the adversary can construct a perfect substitute model matching the architecture and parameters -- but cannot access the per-query randomness of the remote instance. This is strictly stronger than the practical setting (e.g., an adversary does not have a local white-box GPT-5).
>
> 4. In our scheme, the seed is rotated using a random 32-bit unsigned integer. The rotation period determines the real-world time required for a successful nagging attack. For example, under a daily rotation, executing a nagging attack with $n=100$ requires 100 independent days. The model’s accuracy is identical across rotation cycles; only the attack’s feasibility changes.

---

### Official Review · Reviewer_S2rm · 2025-11-03

**Soundness:** 2
**Presentation:** 2
**Contribution:** 2
**Rating:** 2
**Confidence:** 3

**Summary:**

The paper seems to suggest a very board investigation of randomness in deep learning models to state that an adversary with access to an identical target model but not the noise distribution (in the context in which it is used in defence). The paper argues that under this threat model, repeated queries can eventually lead to an adversary discovering a sample to break a defence.

**Strengths:**

- The paper takes a formal analysis to study a problem
- Interesting to consider randomisation methods in deep learning and other than making evaluation of methods harder, to ensure they provide a measurable advantage vs. cost.

**Weaknesses:**

- the idea or randomisation is not new
- the paper has a very simple idea, but the writing is over complicated
- The first two paragraphs in the introduction are, to me, unnecessary and confusing, trying to relate the various uses of noise or randomness to a single question is a huge oversimplification, when it is much more context dependent/should be nuanced. I find the use of the phrase randomized models misleading, it seems to refer to generative models– the introduction is overly verbose for something that is simple to describe without going too board.
- The basic question seems to be very board, but study eventually seems to consider Randomised Smoothing and a denoiser for sanitizing adversarial inputs.
- Changing model parameter is expensive – consider the cost or re-training models, storage costs etc. to a model provider
- From what I can understand, the proposed rotation simply violates the threat model assumed--now the surrogate does not match the target, effectively this is the standard setting used in randomisation methods, but to me this is not shifted out of the standard threat model and instead under the proposed strong threat model shown as a weakness.

**Questions:**

- How many models were trained for each task to select from?
- What is the training cost and storage cost for such a scheme?
- I would be interested in seeing results for more practical task like ImageNet with a state-of-the-art model - for example RS methods scale to ImageNet tasks.
- What is the certification guarantees under random model sampling? (what is an adversarial attack under RS? how is this formalised?)
--The references Wang 2024 and Lucas 2024 do not mention “a nagging attack”, Lines 055 – these prior studies are not based on addressing the so called “nagging attack”, it is unclear how the study can generalise these two studies to prior work on nagging attack defenses. Can this be clarified?
- The literature review to me is somewhat misleading, for example Lucas 2023 does not speak about whether randomisation increase or decrease a defence, it simply compares probabilistic classifiers in comparison to deterministic ones. What am I missing?
- The assumed threat models violate what is typically assumed in practical settings by considering a surrogate that is effectively the target model that is being attacked. So an attack becomes a problem of estimating the noise, or the seed used in a pseudo-random number generator (in some contexts).
- The proposed seed rotation simply violates the assumption of the threat model, i.e, now the surrogate has a different seed to the target, but again, a nagging attack should be able to show that this setting is also vulnerable, i.e. it depends on how many different seeds and over what time the seeds are rotated. Would like to understand what the views on this.

---

> ### Author Response · Authors · 2025-11-27
>
> We sincerely thank the reviewer for the detailed and insightful comments. We address the questions below:
>
> 1. **How many models were trained for each task to select from?**
> All randomization methods we tested required only one trained model; randomness was injected during inference. We used the official pretrained checkpoints released by the original works (Cohen et al., 2019; Salman et al., 2020; Nie et al., 2022). We evaluated all the models with a reproducible seed of 5678. We will release the code for reproducibility upon acceptance.
>
> 2. **What is the training cost and storage cost for such a scheme?**
> For the seed-rotation scheme, there is no extra training cost. Any existing randomization method may be transformed to the seed-rotation scheme without modification on the model. The only extra storage cost is the random seed of the RNG, typically a 32-bit integer. Fixing the random seed for each input ensures no randomization within a short period (the seed rotation cycle).
>
> 3. **RS methods scale to ImageNet tasks:** Seed rotation scheme on ImageNet-1k made a robust accuracy gain of ~2%. Detailed results are provided in Table 2, around Lines 440-442 of the paper.
>
> 4. **Connection to the references Wang 2024 and Lucas 2023**: While they use different terminology, their findings are consistent with the core phenomenon we systematically formalize -- repeatedly querying the randomized model with the same adversarial input leads to significant decrease in accuracy. Wang 2024 phrased it as N-evaluation and resubmit risk (with $N=10$). Lucas 2023 used Nag Factor, from which we adopt the term "nagging". Both studies revealed similar characteristic of randomization methods, and did not directly address the so called "nagging attack" on randomization. We generalise these two studies by formalization of this characteristic of randomization, and propose direct method of addressing the problem with seed rotation.
>
> 5. **Clarification on Lucas 2023**: Correctly noted by the reviewer, Lucas 2023 compares probabilistic classifiers to deterministic ones. Their main conclusion is that deterministic defenses were similarly robust to randomized defenses, but randomization introduced the Nag Factor risk. Therefore, randomized defenses are not as robust as what previous studies expected.
>
> 6. Our evaluation adopts a strong and standard adversarial model: the attacker can build a surrogate that matches the architecture and parameters of the deployed model, but -- critically and naturally -- cannot access the per-query random seed or injected noise. This assumption aligns with the established evaluation protocol for randomized defenses (e.g., Cohen et al., 2019; Lucas et al., 2023; Wang et al., 2024). Under this setting, an attack becomes a problem of estimating the noise, and we show the best achievable performance of such an adversary by exploiting the noise distribution. Section 3 provides additional clarification on this threat model.
>
> 7. Same as above, the adversary does not know the random seed. The adversary may try different seed or exploit the seed distribution, but they cannot directly synchronize with the model’s randomness. In our scheme, the seed is rotated using a random 32-bit (unsigned) integer. The rotation period influence the span of time that a nagging attack needs in this scheme. For example, a daily rotation indicates that the accuracy of nagging attack $n=100$ needs 100 days to achieve in reality.

---

### Meta-Review · Area_Chair_ZdC3 · 2026-01-21

**Summary:**

This paper points out and studies a vulnerability of stochastic defenses to repeated queries (nagging attacks), and introduces a Adversarial Nagging Risk (ANR) metric and a random seed rotation strategy to reduce the effectiveness of these attacks. Reviewers raised concerns about novelty and significance: several viewed the observation that repeated queries increase defense failure probabilities as obvious, questioned the choice of the ANR metric, and questioned the thoroughness of the experimental evaluation. These concerns remained after the rebuttal process. The paper points to a true weakness of stochastic defenses and provides a plausible defense, but its contributions were seen as insufficiently strong to reach acceptance.

**Reviewer Concerns:**

addressed:
- some details of empirical evaluation

remaining issues:
- significance of the problem
- suffficiency of the empirical evaluations
- adequacy of the ANR metric

**Reviewer Scores:**

NA

---

### Decision · Program_Chairs · 2026-01-26

Reject